# Thalamic regulation of frontal interactions in human cognitive flexibility

**Ali Hummos**[1,2☯], **Bin A. Wang**[3,4☯], **Sabrina Drammis**[1,2,5☯], **Michael M. Halassa**[1,2]*, **Burkhard Pleger**[3,4]

**1** McGovern Institute for Brain Research, Massachusetts Institute of Technology, Cambridge, Massachusetts, United States of America, **2** Department of Brain and Cognitive Sciences, Massachusetts Institute of Technology, Cambridge, Massachusetts, United States of America, **3** Department of Neurology, BG University Hospital Bergmannsheil, Ruhr-University Bochum, Bochum, Germany, **4** Collaborative Research Centre 874 "Integration and Representation of Sensory Processes", Ruhr University Bochum, Bochum, Germany, **5** Computer Science & Artificial Intelligence Laboratory, Massachusetts Institute of Technology, Cambridge, Massachusetts, United States of America

☯ These authors contributed equally to this work.
* mhalassa@mit.edu

**Data Availability Statement:** Code made publicly available at <Github.com/hummosa/MD-reservoir>. Behavioral data and group-level fMRI data available at <https://ruhr-uni-bochum.sciebo.de/s/xKBPyW7ZGLs2q2g>.

## Abstract

Interactions across frontal cortex are critical for cognition. Animal studies suggest a role for mediodorsal thalamus (MD) in these interactions, but the computations performed and direct relevance to human decision making are unclear. Here, inspired by animal work, we extended a neural model of an executive frontal-MD network and trained it on a human decision-making task for which neuroimaging data were collected. Using a biologically-plausible learning rule, we found that the model MD thalamus compressed its cortical inputs (dorsolateral prefrontal cortex, dlPFC) underlying stimulus-response representations. Through direct feedback to dlPFC, this thalamic operation efficiently partitioned cortical activity patterns and enhanced task switching across different contingencies. To account for interactions with other frontal regions, we expanded the model to compute higher-order strategy signals outside dlPFC, and found that the MD offered a more efficient route for such signals to switch dlPFC activity patterns. Human fMRI data provided evidence that the MD engaged in feedback to dlPFC, and had a role in routing orbitofrontal cortex inputs when subjects switched behavioral strategy. Collectively, our findings contribute to the emerging evidence for thalamic regulation of frontal interactions in the human brain.

## Author summary

The expansion of frontal cortex during mammalian evolution suggested a prominent role in intelligent and adaptive behavior, overshadowing earlier parts of the brain. However, recent rodent studies have pointed to a role for the cognitive mediodorsal thalamus (MD) in sustaining and flexibly switching representations in the frontal cortex, but direct relevance to human decision-making are unclear. Here, inspired by animal work, we extended a neural model of an executive frontal-MD network and trained it on a human decision-making task for which human neuroimaging data were collected. We found that

**Funding:** SD is funded by National Science Foundation awards CCF-2139936, CCF-1810758, CCF-0939370, and CCF-2003830. The funders had no role in study design, data collection and analysis, decision to publish, or preparation of the manuscript.

the model MD thalamus learned an abstract representation of its cortical inputs and provided direct feedback to frontal cortex leading to flexible computations and enhanced task switching. These abstract MD representations and ability to re-organize frontal computations created an efficient mechanism where MD can integrate input from other regions to select behavioral strategy dynamically. The model predicted an efficient route through the MD for frontal region interactions and we found consistent evidence in human neuroimaging data. Collectively, our findings contribute to the emerging evidence for thalamic regulation of frontal interactions in the human brain.

## Introduction

The expansion of frontal cortex during mammalian evolution is thought to have given rise to advanced cognition [1]. In humans, frontal cortex consists of several regions that are generally thought to have specialized functions [2–5]. For example, the dorsolateral region is linked to executing cognitive functions [6], while the orbitofrontal cortex (OFC) is associated with adaptive and flexible responding in a changing environment [7]. The ventromedial region is associated with estimating the value and salience of sensory and cognitive variables and thereby, guides value-based decision-making [8].

Interactions between these regions are thought to implement a variety of functions relevant to the flexibility by which cognitive resources are deployed. For example, interaction between medial prefrontal cortical regions and the anterior cingulate cortex are thought to implement a process of updating beliefs about higher-order contextual associations [9–11]. Interactions between OFC and dorsolateral prefrontal cortex (dlPFC) are thought to implement a form of reinforcement learning [12,13], which has been recently modeled by artificial recurrent neural networks (RNNs) [14].

Recent studies have indicated a role for the thalamus in cortico-cortical interactions in non-human animals [15,16] and humans [17,18]. These studies have provided a complementary perspective on thalamic function to the classical notion of a relay, often seen in thalamic regions receiving inputs from sensors and projecting to primary sensory cortex [19]. Specifically, such studies of associative thalamic regions have indicated a role for this subcortical collection of excitatory neurons, devoid of recurrent connections, in sustaining, switching, and synchronizing connected cortical areas [15,20–24]. For example, in mice, multiple studies have shown that the mediodorsal thalamus (MD) projects to areas of prefrontal cortex with signals able to sustain or switch task-relevant activity patterns, enabling the maintenance of working memory on one end and task switching on another [20,21]. The anatomical divergent and convergent cortical projections originating from the thalamus makes the thalamus an ideal location not only for information integration but also deployment of simultaneous instructions [25,26]. Specifically, studies have shown diffusive projections from neurons in MD thalamus across several areas of prefrontal cortex in rodents [27–29,29,30]. Beyond the MD's dense connections with frontal cortical regions, the structure also receives projections from regions within the temporal cortex, the midbrain, and the basal ganglia [16,27,31], positioning it well to have a broad integrative function. Given the studies above, two questions are pertinent: first, is the MD thalamus a functional mediator and a communication bridge for frontal cortical areas? Second, what are the precise computational functions of such engagement?

In this study, we built on recent work in rodents and extended a neural model that captures interactions between the MD and frontal cortex [21], hereafter referred to as the

thalamocortical "neural model". We implemented biologically plausible learning rules at the corticothalamic projections capable of learning abstractions of the dominant representations in dlPFC (as inferred from the rodent work). A key model assumption from the rodent work was that the MD thalamus performs an intermediate-level computation rather than a sensory signal relay [20,21]. We first confirmed that this is the case for the human brain, by examining fMRI data of human participants performing a probabilistic inference task [32]. We then asked the neural model to solve components of the same probabilistic inference task and identified a role for the MD in rapid and flexible gating of human dlPFC strategy representations.

We next used the model to interrogate computational mechanisms underlying interactions across frontal regions. Therefore, we considered computations partly attributed to the OFC in representing latent states in a task [33,34] and appropriately switching them at change points [35]. We considered whether an OFC model would communicate its representation of behavioral strategy using a corticocortical OFC-dlPFC pathway or using a transthalamic OFC-MD-dlPFC pathway. Simulations showed that leveraging the thalamic route by OFC inputs required far fewer neurons and a shorter-lasting switch signal compared to a direct OFC-dlPFC pathway. Critically, human fMRI data subjected to dynamic causal modelling was consistent with the trans-thalamic route, providing evidence for the MD having a central role in regulating distributed frontal cortical interactions. All told, our modeling and experimental analysis provide compelling evidence for thalamic gating and integration in human cognitive flexibility.

## Results

### Thalamocortical neural model solves human task and shows comparable behavioral dynamics

**Human probabilistic inference task.** Participants learned the predictive strength of a tactile cue (up or down) in forecasting a subsequently presented target stimulus (up or down, Fig 1A) [32,36]. Participants responded by either matching the input cue (match rule) or responding with the opposite direction (non-match rule). The task consisted of unannounced pseudo-random blocks with different association levels between rule and rewards, which we refer to as distinct contexts requiring different sensorimotor mappings akin to the rodent work [21]. Contexts had cue-reward associations that were strongly predictive (90% of match responses rewarded, or 10%), moderately predictive (70% and 30%) and non-predictive (50%).

**The neural model.** Previous data analysis of fMRI signals in this paradigm revealed multiple interactions between frontal cortical areas and the MD thalamus [32, 37]. To derive insight into putative computational mechanisms, we extended a neural model encompassing the MD and an executive PFC region (Fig 1B) which had been used to implement a context switching task in mice [21]. Our neural model includes a reservoir of recurrently connected neurons to model the executive prefrontal cortex (dlPFC in primates), consistent with prior modeling efforts in this domain [38,39]. Sensory input (up and down tactile cues) are projected onto the dlPFC (presumably through sensory cortical hierarchy), consistent with recordings of sensory responsive neurons in the mouse prefrontal cortex [21], and analysis of human fMRI detailed below. Responses (up and down) are read out from two output units with the output weights learned through a biologically-plausible learning rule (node perturbation [40], see Methods).

Model MD neurons are devoid of local excitatory connections consistent with electrophysiological findings [41] and include a winner-take-all mechanism representing mutual inhibition through the thalamic reticular nucleus [41,42]. Previous work in mouse indicated that the MD thalamus integrates its prefrontal inputs to generate a compressed representation of the task's context [20,21]. We explored whether such representation could emerge in the model

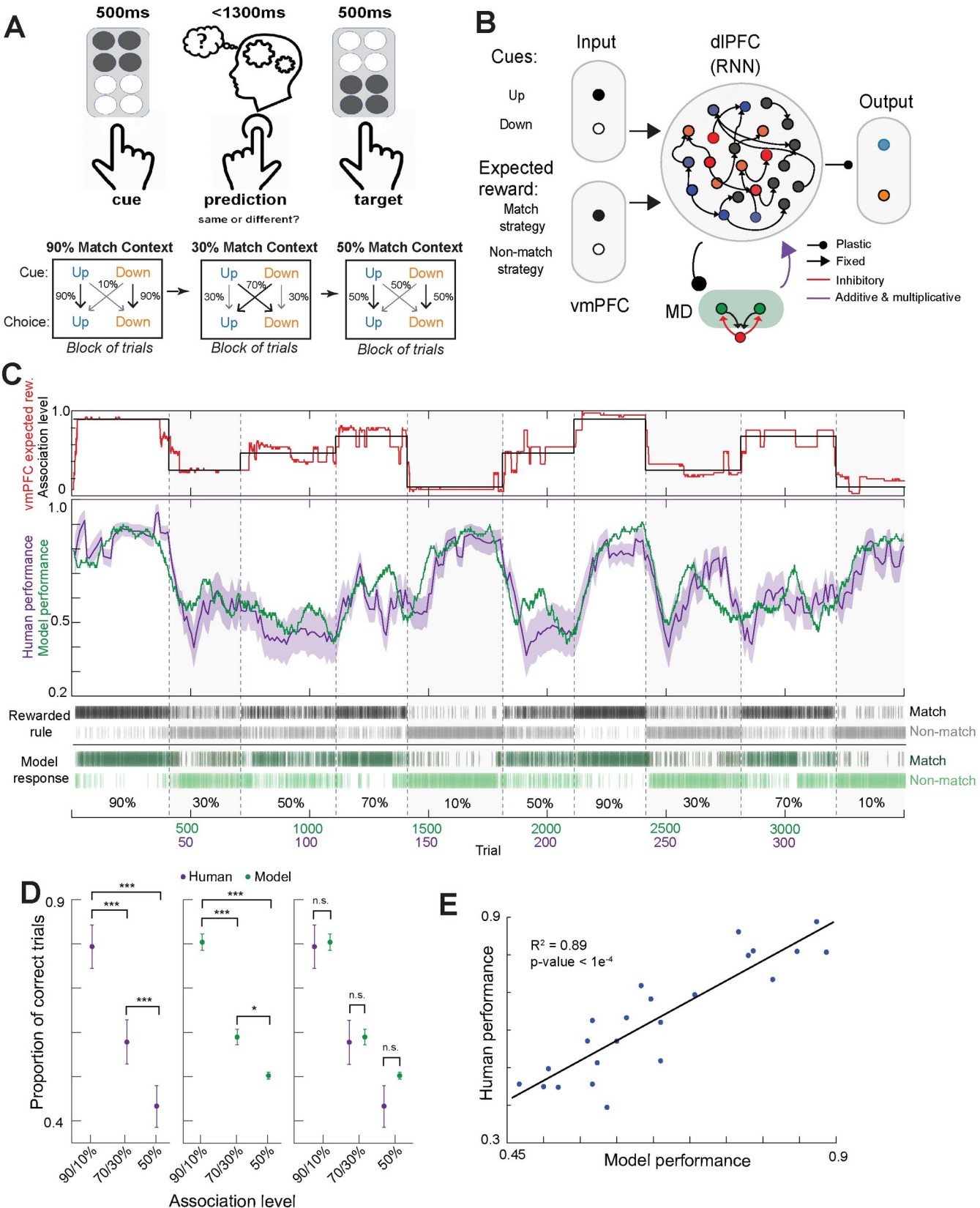

**Fig 1. Comparison between human participants and neural model performance on a probabilistic inference task. A.** Behavioral task design: on individual trials, human participants were asked to generate a behavioral choice based on a somatosensory cue input. Inputs were either up or down, and the mapping onto the choice (match or non-match) changed covertly across blocks of ~50 trials. Participants were told what the correct choice was on every trial (target). The mapping was probabilistic with different association levels (90%, 70% and 50%). **B.** Cartoon of the main neural model used in this study. The model contains a recurrently connected reservoir of rate neurons that capture the dlPFC, with long range connections to feedforward MD neurons with a winner-take-all mechanism. The model receives task relevant sensory inputs (up and down) as well as value of the mapping strategies (match and non-match). The model outputs behavioral choice (up or down) through two output neurons that linearly decode the reservoir activity. **C.** Mean task performance of human participants (purple) and of the neural model (green) plotted over 10 blocks of association levels, to show qualitative similarity (middle). Grey bars reflect the correct rule that was rewarded on that trial and green bars show the model response (bottom). The red trace shows the match strategy value signaled by the vmPFC (top, see Methods) which served as an input to the dlPFC RNN. These expected reward values closely follow the association levels through the experiment (black line, top). **D.** Comparison of model and human performance of high (90/10% predictability), low (70/30%), and non-predictive (50%) association levels. Pairwise comparisons of human responses across association levels were statistically significant for all pairs (left; two-way ANOVA; 90/10% and 70/30%, ***$p < 0.0001$; 90/10% and 50%, ***$p < 0.0001$; 70/30% and 50%, ***$p < 0.0001$). Pairwise comparisons of model responses across association levels were statistically significant for all pairs (middle; two-way ANOVA; 90/10% and 70/30%, ***$p < 0.0001$; 90/10% and 50%, ***$p < 0.0001$; 70/30% and 50%, *$p = 0.039$). Comparison of model and human responses with association levels revealed no significant differences for all association levels (right; two-way ANOVA). **E.** Correlation between human and model performance with randomly sampled chunks of trials (humans: 15 trials, model: 150 trials) sampled from all trials (Pearson correlation, $R^2 = 0.89$, ***$p < 0.0001$).

based on a Hebbian learning rule at the dlPFC to MD corticothalamic connections, inspired by recent work [43] suggesting that Hebbian learning mediated unsupervised clustering. Using higher number of MD neurons showed similar results so for simplicity, we kept the number at 2 neurons for the current study.

Previous experimental evidence showed that MD thalamic projections engaged direct inhibition of prefrontal neural activity and indirect amplification of local functional connectivity through disinhibitory mechanisms [20,44,45]. We model these experimental observations by an additive (suppressive) and multiplicative (amplifying) thalamic output to the prefrontal reservoir recurrent neural network (RNN). Model behavior did not require learning at these projections and they were chosen randomly and held fixed for stability.

A key difference between the previous animal work upon which our neural model was designed and the current task is that inferring a block switch in the current task requires a record of recent rewards. So, we provided the model with inputs representing the value of the two behavioral strategies in recent trials (i.e., expected rewards from match vs. non-match strategies). These value inputs improved model behavioral flexibility and provided a signal to disambiguate the two strategies based on recent reward patterns (S1A–S1D Fig). Consistent with this modeling assumption, fMRI data of humans performing the same task showed that vmPFC activity correlated with prior belief about outcome value, indicating the availability of such signals in the prefrontal region (S1E and S1F Fig, also see [46,47]).

**Human and model performances.** Our dlPFC-MD neural model was able to solve the human inference task, exhibited performance qualitatively similar to human performance, and was able to flexibly switch between the behavioral strategies at block changepoints (Fig 1C–1E). Specifically, the neural model was able to match human performance on outputting the correct rule over blocks with multiple association levels and with qualitatively similar dynamics in switching between blocks (example in Fig 1C). Human participants learned the task successfully, with more correct performance in blocks with high predictability (90%/10% association levels) compared to low predictability (70%/30%) or unpredictability (50%; Fig 1D, left). There were no significant differences in behavioral performance between human data and the neural model when sampled from high, low, and non-predictive association levels (Fig 1D); performance was highly correlated when randomly sampled from trials across the task (Fig 1E).

## Human fMRI reveals MD engagement in a cognitive role in the task

To draw direct parallels between the neural model and human data, we returned to our previously collected dataset [32] and took a different analytical approach from the published work.

In mice, prefrontal inhibition impaired overall performance, while MD inhibition led to impairment specifically when animals were required to switch behavior [21]. As such, we constrained the generalized linear model (GLM) analysis to the prefrontal and thalamic areas, and used a smaller smoothing kernel for improved spatial resolution (see Methods). With this new analysis, we were able to directly evaluate the engagement of different prefrontal and thalamic areas by a small-volume correction [48] (for brain-wide GLM see S1 and S2 Tables). Within the frontal regions, GLM showed significant activity modulation in dlPFC in trials involving strategy switching (*Switching>Staying*) ((t (27) = 7.07, p < 0.001; small-volume FWE-corrected; Fig 2A), and interestingly, also the orbitofrontal cortex (OFC) (*Switching>Staying*, t (27) = 7.31, p < 0.001; small-volume FWE-corrected, Fig 2A, middle).

Within the thalamic region, the MD thalamus was the only thalamic area modulated by switching (Fig 2A, right, peak MNI coordinates x/y/z = 12/−10/8, t (27) = 4.37, p = 0.02, small-volume FWE-corrected). These findings corroborated the involvement of these areas included in the neural model as a direct interpretation of physiological recordings from mice performing a similar task [21], and added the OFC as a region of interest which we introduce to the model in a later section.

This frontal network switching motif (dlPFC, OFC and MD) offered a starting point for testing one of the key neural model assumptions; that the MD thalamus is engaged in an intermediate computational role (compressing dlPFC sensorimotor mapping) rather than as a relay of sensory inputs to the dlPFC. By leveraging Dynamic Causal Modelling (DCM) together with family-level Bayesian model selection (BMS) on fMRI data, we first evaluated three competing hypotheses; do sensory tactile inputs arrive to the dlPFC, OFC, or MD (DCM C-matrix, Fig 2B)? BMS evaluates the evidence that compromises the accuracy and complexity of a causal model and uses the group log-evidence (which is equivalent to the log group Bayes factor) to quantify the relative goodness of fit. The sum of the log-evidence over subjects can be used to compute how likely it is that a specific model generated the data at the group level (i.e., the expected model posterior probability). Consistent with the neural model construction and a direct extrapolation from previous animal work [21], we found that the model posterior probability revealed very strong evidence in favor of the causal model in which sensory inputs first enter the dlPFC-MD-OFC circuit through the dlPFC (posterior probability = 1.00, Fig 2C). While the non-relay functions of thalamic nuclei in the human brain are suspected, evidence remains indirect and, to our knowledge, this is one of the few direct demonstrations that a human thalamic area engages in a task not to relay sensory inputs. In addition, this finding also relates to controversies about the direction of communication between the dlPFC and OFC where some evidence points to the OFC as the recipient of multi-modal sensory input [49] and thereby, likely forwarding such information to dlPFC, while other evidence points to OFC being hierarchically higher and contextualizing representations in dlPFC [50]. Our findings are in support of the later view, at least in this task. Our analysis did not include lower-level sensory areas but focused on the subgraph of the three areas implicated in cognitive flexibility in this task.

## Neural model MD activity suggests a role in flexible dlPFC switching

To gain computational insight into our neural model of dlPFC-MD, we considered the encoding properties of dlPFC and MD. As predicted from previous studies [21], the dlPFC reservoir showed mixed selectivity, with neurons encoding a variety of response patterns (Fig 3A–3C), including neurons encoding a cue in a single context (Fig 3A), neurons encoding the same cue across contexts (Fig 3B) and neurons switching their encoding across contexts (Fig 3C). Importantly, those neural patterns were readily read-out as behavioral responses in the output

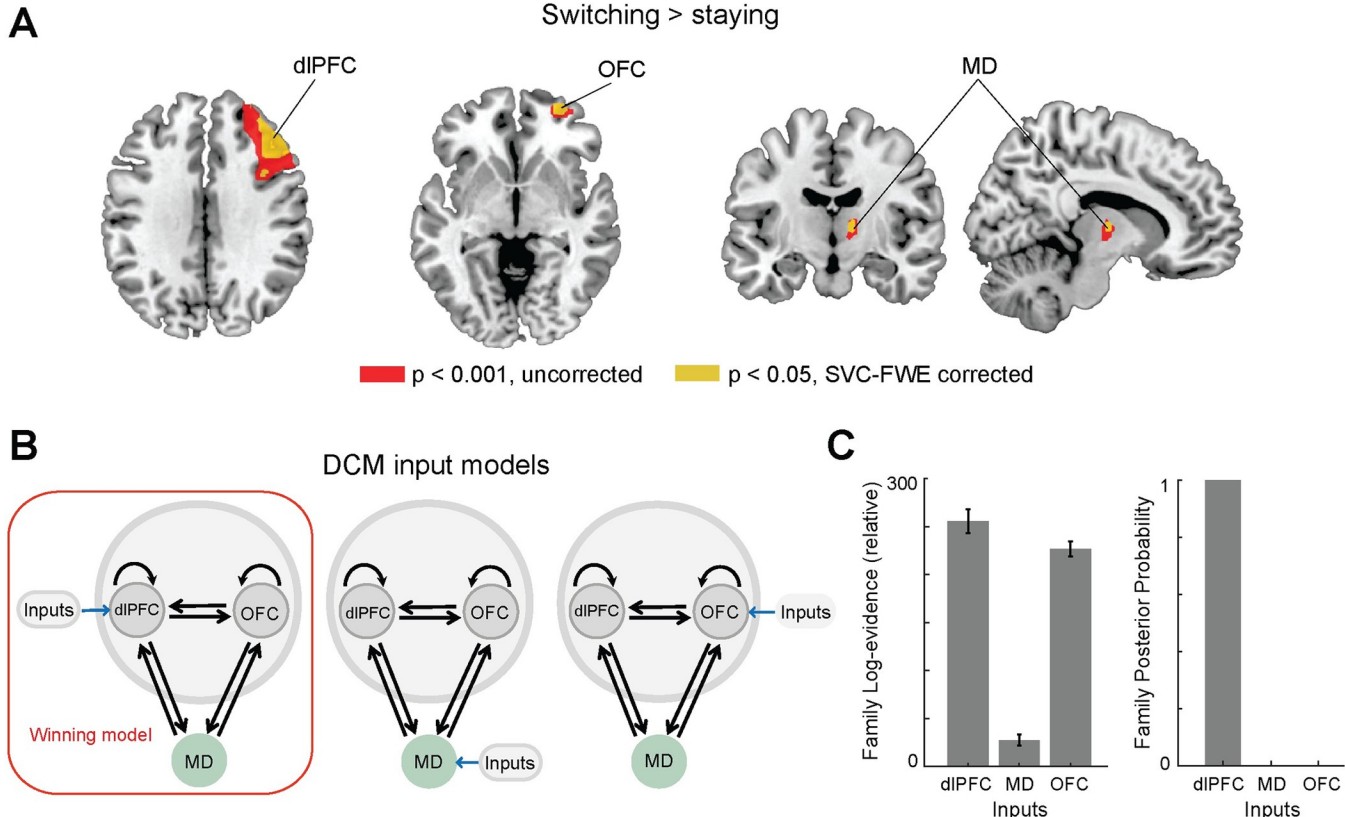

**Fig 2. Human fMRI confirms the dlPFC, not OFC or MD, receives the driving sensory input. A.** Strategy switching (*Switching>Staying*) entailed significant activity in right dlPFC, right OFC and right MD; here projected on axial, coronal and sagittal MRI brain slices. Brain activations displayed at p < 0.001 (uncorrected, red) and p < 0.05 (whole brain FWE correction, yellow; small-volume correction, blue). **B.** Models with dlPFC, MD and OFC receiving sensory tactile driving "inputs" respectively. **C.** Bayesian Model Selection revealed dlPFC, not OFC or MD, as the input region. The left panel shows the average and standard error of log-evidence for all models in the corresponding inputs family (we have 6 models for each input family, see S5 Fig). The sum of the log-evidence over subjects was used to compute the expected model posterior probabilities of each input family which are shown in right panel. (dlPFC–dorsolateral prefrontal cortex; OFC–orbitofrontal cortex; MD–thalamic mediodorsal nucleus).

neurons (up versus down; S2A and S2B Fig). In comparison, neural model MD neurons encoded the dominant strategy within a block, which we interpret as a temporal context, as it is not signaled by a contextual cue but rather can be inferred from its structure in time (Fig 3D and 3E). MD neurons signaled the temporal context by exhibiting a dominant output that co-varied with the level of association across blocks (S2C and S2D Fig).

We considered how these model MD representations might emerge and examined the Hebbian learning dynamics at the dlPFC-MD corticothalamic connections. Since components of Hebbian plasticity operated at different timescales [51–53], we considered a dynamic Hebbian eligibility trace and parametrized its time-constant. Smaller time-constant values biased MD to encode the fast-changing signals (up or down cues) while larger time-constant values biased MD to encode the slower temporal context signals (i.e., encoding the dominant strategy within a block, Fig 3F). This finding suggested that the brain may be using long-timescale eligibility traces at the corticothalamic connections for the thalamus to encode current temporal context.

To evaluate the contribution of MD activity patterns to dlPFC function, we performed a perturbation study in which we eliminated the output of the MD to the dlPFC. Compared to the MD intact model, the MD lesioned model exhibited fairly equivalent steady-state performance but had perturbed transient performance when the level of association switched across

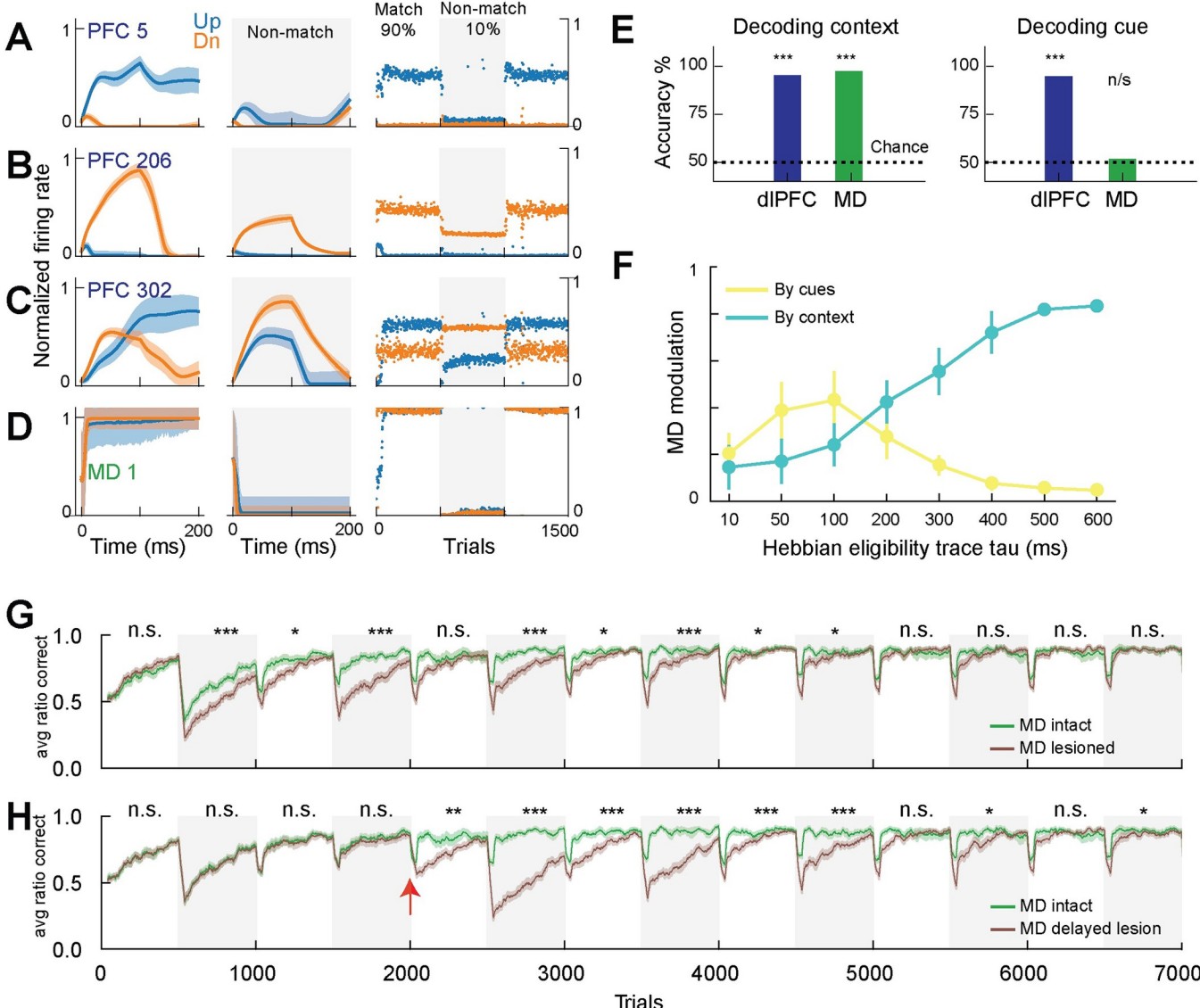

**Fig 3. Context encoding in MD enhances prefrontal cognitive flexibility in the neural model. A-C.** Example responses from 3 dlPFC neurons to sensory input ('Up' or 'Down') in trials where match strategy was rewarded (match block, left panels), or non-match strategy was rewarded (non-match block, middle panels), and the trial averaged responses over the first 3 blocks of the experiment (right panels). **D.** The same panels for one MD neuron showing responses selective to match context. **E.** Quantification of the coding properties using regression to decode task-relevant variables from population activity of either dlPFC or MD. Context could be decoded from either dlPFC or MD (left), but cue was only decodable from dlPFC activity and not MD (right). **F.** We tested the modulation of MD activity by rule or by cues by taking the correlation between trial-averaged MD activity and a vector representing either the cues (1 for up trials and -1 for down) or rule (1 for match rule or -1 for non-match). To demonstrate the effect of the dynamic Hebbian eligibility trace we tested MD modulation as we varied the eligibility trace time constant (tau). Lower (higher) tau values biased MD to encode sensory cues (rule). **G.** Comparing mean (± STD) of ratio correct responses for MD intact and MD lesioned models from 20 random instantiations of the models. Experiment had blocks alternating between 90% and 10% match trials as other association levels showed similar patterns. **H.** Same as in G, but showing the effects of lesioning MD after training with MD intact for a few blocks (lesion marked with red arrow). Statistical testing with t-test over performance means in each block for 20 random runs.

blocks (switching deficit; Fig 3G). We noticed similar patterns with other association levels (70%, 50%, 30%) and we only show alternating 90% and 10% blocks in following experiments for clarity. Of note, the difference between the MD intact and MD lesioned models starts to vanish, indicating that after multiple rehearsals the cortical network may eventually learn the two rules and switch between them using slow synaptic plasticity. Further, in a model trained

with intact MD, lesioning after acquisition of behavior causes a similar pattern with poor performance selectively in trials following rule switches, but again the plasticity from dlPFC to the output layer eventually recovers the behavior with sufficient training (Fig 3H). These impairment patterns are consistent with animal experiments where MD inhibition delayed the acquisition of tasks requiring flexible responses [23,24,54–56], but eventually MD inhibited animals achieved normal performance after extensive training [23,24,57].

To understand the effect of MD engagement on dlPFC encoding of task-relevant variables, we systematically compared dlPFC neural activity patterns across the MD intact and MD lesioned neural models. We found that MD engagement reduced the number of dlPFC neurons that show cue selectivity (up vs. down) across the two behavioral contexts rather than only in one (Figs 4A and S3), indicating that the MD may impart a higher degree of computational specialization within the dlPFC. Consistent with this notion, the connection weights from cue selective neurons in the dlPFC reservoir to the output neurons grew more consistently during the appropriate behavioral context, if they were cue selective only in that context (Fig 4B and 4C, see Methods), indicating that MD partitioning of dlPFC activity patterns allowed for readily separable and computationally efficient representations in dlPFC.

We sought to dissociate the contributions of the multiplicative and the additive thalamocortical projections, representing, respectively, the activation of fast spiking and the indirect inhibition of dendrite-targeting dlPFC interneurons [58]. We varied the strength of each type of projections by multiplying the corresponding weights by a factor. We noticed that increasing the weights of the additive projections increased performance and cognitive flexibility up to a limit, after which performance deteriorated again (S4A and S4C Fig). Performance increase coincided with decreased correlation of neural activity between match trials and non-match trials (S4C and S4E Fig), while deterioration in performance coincided with increased correlation between the up and down trials, decreasing the separability of incoming sensory inputs and impairing sensory processing (S4E Fig). In contrast, gradually amplifying the multiplicative effect improved performance significantly, reaching the ideal ceiling performance after learning the behavior in the first two blocks (0.9 ratio correct from the $3^{rd}$ block onwards) (S4B and S4D Fig). Increasing the multiplicative projection strength lowered the neural activity correlation between the two contexts while maintaining low correlation between the two sensory cues (S4F Fig). Of note, to isolate the contribution of the thalamocortical projections, this experiment had minimal model components with one MD neuron artificially activated for each context (no corticothalamic learning), and no value inputs (from our representation of vmPFC).

## Model MD provides an efficient route for dlPFC-OFC interaction in strategy switching

Analysis of human prefrontal fMRI revealed engagement of OFC in the network switching motif (dlPFC, OFC and MD), therefore we extended our model with a module representing OFC computations, to gain further computational insight into the role of the MD in possible OFC-dlPFC interactions. The OFC is known to represent latent states in a task [33,34] and switch them appropriately at change points [35]. Therefore, such latent variables could adjust dlPFC activity patterns beyond what the MD alone may provide. To test this hypothesis with minimal further assumptions on the neural architectures involved, we abstracted OFC as two nodes representing inference over the current block, one node representing predominantly match strategy blocks and the other, non-match strategy blocks. The two nodes were updated according to a Bayesian estimator of the probability of an experiment block switch, which enabled the OFC representation to detect block changes faster than the dlPFC-MD circuit

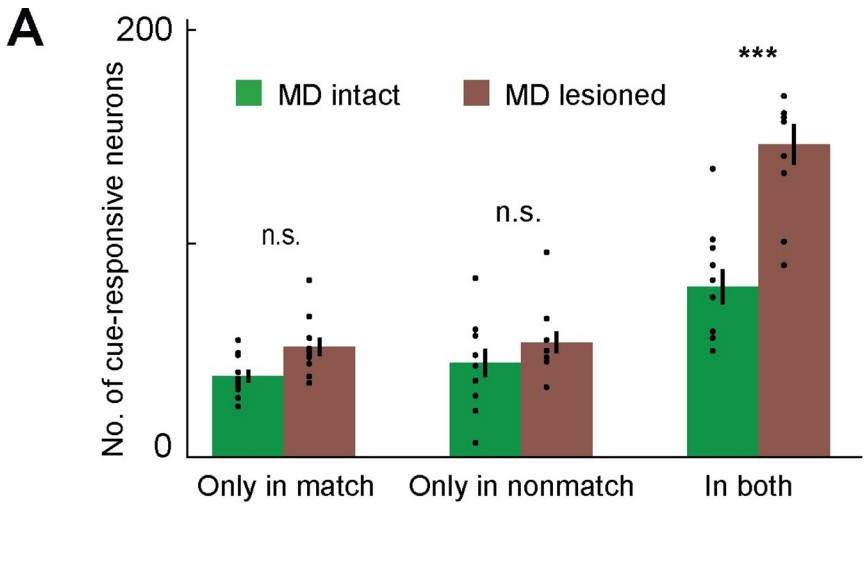

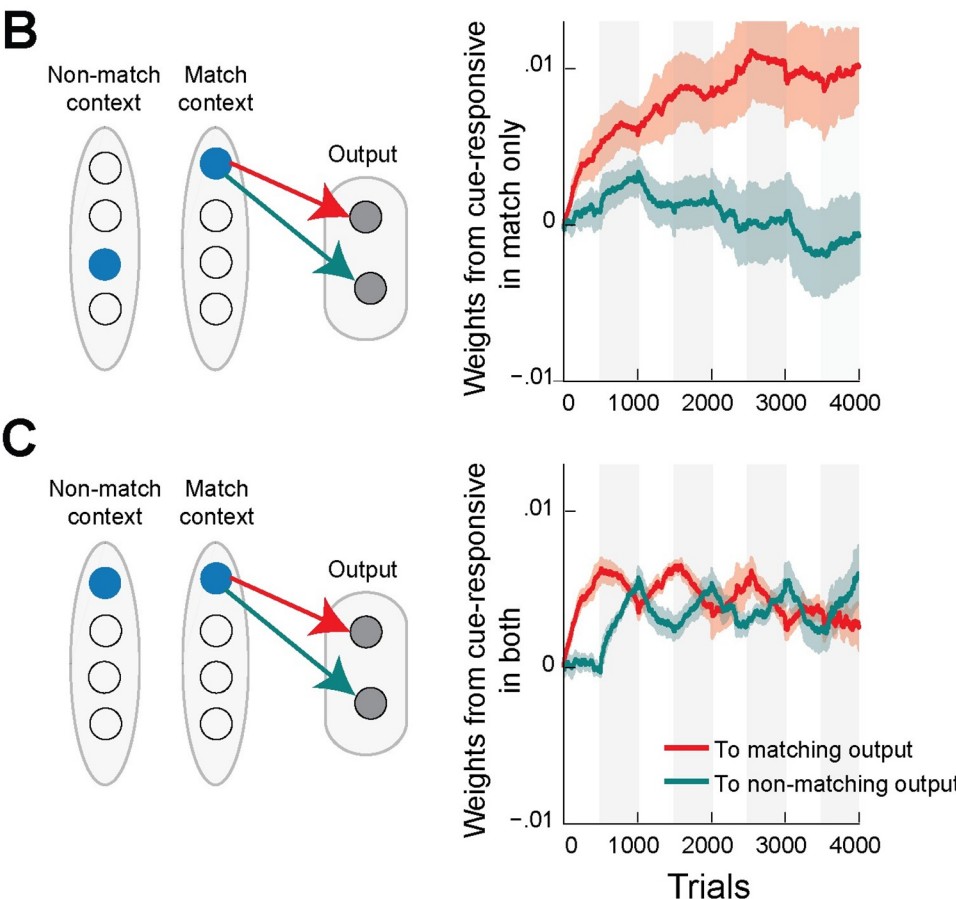

**Fig 4. Selective dlPFC sensory representations in the MD intact neural model. A.** We identified model dlPFC cells that were cue-responsive in one of the two behavioral contexts selectively and cells that were cue-responsive in both contexts non-selectively, using logistic regression. We found the most significant difference between the MD intact and MD lesioned models was an increase in non-selective cells with MD lesioning (Mann-Whitney U test, two-tailed, both contexts p-value 0.003, match-selective p = 0.210, non-match selective p = 0.449). Bars represent means ±SEM **B.** Schematic shows cartoon of cells that were cue-responsive only in one of the behavioral contexts, and the connections to the appropriate output cell (left). These cells had their weights to the appropriate output neuron increase coherently in the corresponding behavioral context and remain dormant out of context (right). **C.** In comparison, cells that

responded to the same cue in either context showed impaired learning with forgetting of previous learning after each experimental block, as rewarded behavioral responses reversed. We show weight averages from the population of dlPFC cells encoding on of the input cues, as identified by logistic regression, to the appropriate output neuron, with shaded areas representing ±SD.

alone could. Hebbian synapses connected the two OFC nodes to MD and dlPFC neurons, allowing the OFC representation to learn which MD and dlPFC neurons were active for each behavioral strategy.

As such, the OFC can now use its representations and learned connections to the neural model to cast a vote on current behavioral strategy. We considered two possible hypotheses, that OFC would send its vote directly to dlPFC (cortico-cortical pathway), or through MD (transthalamic pathway). We found both pathways to be effective at improving performance in the trials right after a block switch (Fig 5A and 5B), as the OFC Bayesian estimator rapidly detected the switch and biased the dlPFC-MD circuit towards the appropriate behavioral strategy. However, the pathway to dlPFC required learning 1000 parameters and sending the signal to 500 dlPFC neurons. To compare the computational efficiency of the two pathways, we explored whether the number of involved dlPFC neurons can be lowered, but we found that performance deteriorated rather rapidly with numbers less than the full 500 neurons (Fig 5C). In contrast, the MD pathway required learning 4 parameters and sending the signal to only 1 of the 2 MD neurons (Fig 5C). While this result depends to an extent on our proposal that MD has a more compressed representation of the task context, these trends would also hold due to MD having fewer number of neurons. In addition, the MD pathway required the OFC signal to be active for fewer timesteps at the beginning of each trial (Fig 5D). As such, the model indicated computational advantages to the transthalamic pathway, and we next turned to human data to find support for either pathway in the human brain.

## Human fMRI consistent with a transthalamic pathway for switching behavioral strategy

We next applied DCM to human fMRI data and compared resulting patterns of causal interactions amongst dlPFC, OFC, and MD, as humans switched their behavioral strategy (DCM B-matrix). Among six competing dlPFC-OFC-MD causal patterns (S5 Fig) in which the driving input was directed to dlPFC, the pattern with a causal connection from OFC to MD outperformed the alternatives including causal patterns with OFC to dlPFC connections (posterior probability = 0.99, Fig 5E). The Bayesian parameter averaging (BPA) across participants revealed that connection from OFC to MD and from dlPFC to OFC, as well as connections between dlPFC and MD in both directions were all significantly strengthened by strategy switching (*Switching>Staying*, posterior density > 0.95) (Fig 5F). These DCM results based on human fMRI data support the engagement of transthalamic pathway when humans switched their decision strategy.

## Discussion

Altogether, by combining neural network modeling with experimental approaches, our neural model bridged insights from animal recordings to making interpretative predictions about networks of the human brain. The ability to use existing fMRI datasets from human participants was greatly beneficial to guiding and validating the modeling effort.

Our neural model extends previous neural network models of frontal-thalamic networks with the aim of capturing task-relevant computations in the human brain. We introduced Hebbian plasticity at the corticothalamic connections from dlPFC to MD to cluster recent

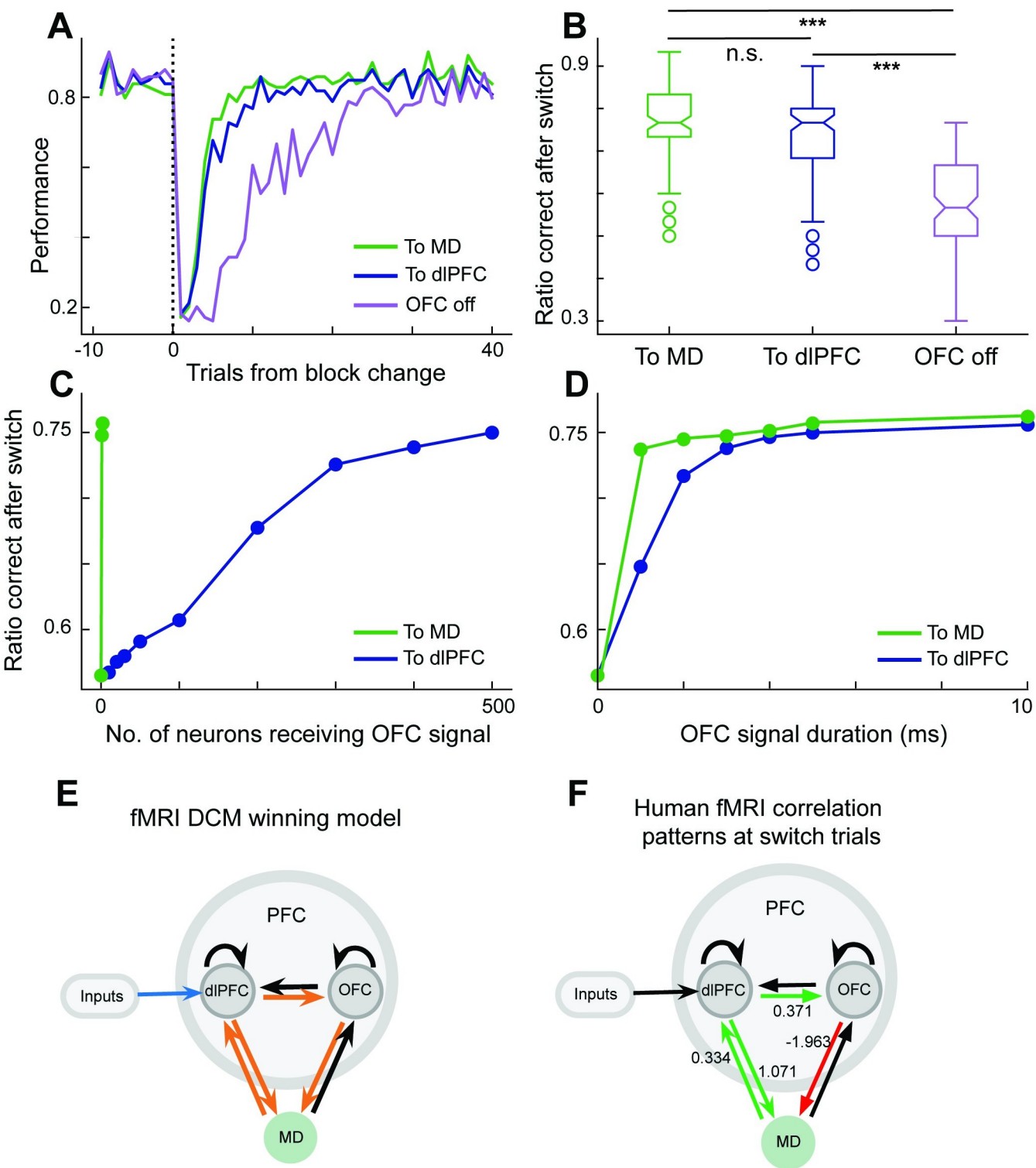

**Fig 5. Human fMRI confirms computational advantage of transthalamic routing of cortical information suggested by neural model.** We considered a Bayesian model capturing computations in the OFC that detected context changes and cued the dlPFC-MD neural model to switch behavioral strategy accordingly. Two nodes of the Bayesian model represented belief in current context (predominantly match or non-match context). Context nodes associated through Hebbian learning to neurons in MD or dlPFC that happened to be active. When the OFC model detected a switch, it activated the appropriate node for the inferred context, thereby activating dlPFC or MD neurons associated with it. **A.** The OFC switch signal enhanced performance immediately after a block change, and both sending the OFC signal to dlPFC or through a transthalamic route to MD performed comparably. **B.** Average performance in the 30

trials after a switch shows comparable enhancement in performance in the model with OFC sending switch signal to MD or to dlPFC, compared with model with no OFC signaling (paired t-test, MD vs dlPFC p = 0.6, dlPFC vs OFC off ***p <0.0001, and MD vs OFC off ***p <0.0001). **C.** The transthalamic pathway required OFC to activate only the 1 of the 2 MD neurons, whereas lowering the number of dlPFC neurons receiving signal immediately deteriorated performance. **D.** We varied the number of timesteps within a trial that OFC input needed to be active, and the transthalamic path required shorter duration signal to achieve ceiling performance. **E.** Turning to human fMRI analysis, Bayesian model selection from fMRI data revealed that the causal pattern with connection from OFC to MD rather than to dlPFC was superior in explaining the difference between trials where humans *switched* strategy or *stayed* in the same strategy (model posterior probability = 0.99, for all model comparisons see S5B Fig). **F.** Modulation of connectivity in *Switching* from fMRI data. Green connections represent positive modulation, red connections represent negative modulation. The number alongside the connections indicate the coupling parameters from Bayesian parameter averaging, which represents the strength of effective connectivity in Hz.

experiences into contexts. The Hebbian plasticity trace evolved over time and served as an inductive bias to detect temporal patterns, similar to recent work [43]. By allowing the network to discover consistent patterns of neural activity, it no longer required the task identifier input provided to MD, as used in previous work [21,59,60]. Analysis of human fMRI data confirmed the active involvement of MD showing distinct patterns of directional interactions when behavioral strategy was switched.

While this probabilistic inference task might be a simplified version of the probabilistic inference humans are capable of, its simplicity allowed for direct comparison with a closely related contextual behavior task in mice [20,45,61]. This setup allowed us to assume that the model architecture inspired by mouse physiological recordings is applicable to the human task, and then verify these assumptions by analyzing human fMRI data. In addition, in comparison to the mouse cross-modal attention task, upon which the model was based, the human task was a purely tactile task, whereas the mouse task is cross-modal, requiring the animals to attend to auditory and visual cues. In the mouse task, the current rule is directly signaled per trial and requires no inference, while in the human task, rules were inferred from monitoring of expected versus actual outcome. Further, humans were instructed that rule changes may occur (although unannounced), whereas animals had to learn the task structure from trial-by-trial errors. Despite these differences, we found that the extended rodent model reliably captured human behavior, emphasizing the robustness of our model.

## Multiplicative gain control reduces catastrophic forgetting

Recent studies found that MD input modulated the gain of cortical recurrent connections multiplicatively [20], and such findings were the basis for a recent neural network model of the thalamocortical loop [21]. This current study extends this work by allowing for plastic corticothalamic connections, and random and fixed thalamocortical connections, adding to the generality of this neural model. Recent modeling work showed a role for multiplicative gating in continual learning problems where it reduces interferences between learned memories and reduces catastrophic forgetting [62,63]. Similar to our method, these models learned to infer task or context boundaries from current inputs, but had a gating mechanism abstracted at a different hierarchical level. For example, the dendritic gated networks mapped the molecular layer interneurons in the cerebellum gating the dendrites of a single Purkinje cell [63], whereas our formulation examines a more coordinated and far-reaching control of the entire dlPFC RNN to redefine its computations dynamically, presumably through coordinated MD modulation of dlPFC local interneurons [20,58].

## Transthalamic communication as a workspace and a coordination hub

Higher order thalamic nuclei provide an alternative transthalamic route to the direct cortico-cortical communication pathway [19,64], but the possible roles of the transthalamic route remain unclear. Here we demonstrate through a neural model and human fMRI analysis that

the transthalamic route can be an efficient pathway to collect votes from prefrontal regions to coordinate current behavioral strategy. While this emphasizes a role for MD as a shared workspace, it is important to note that these contextual signals emerged from interacting with executive dlPFC which also relied on these contextual signals from MD to separate learning of two different input-output transformations. By interacting intimately with MD, dlPFC gained additional flexibility in its computations and more contextualized sensory representations. At the same time, this creates a structure where dlPFC computation can be modulated, by other brain areas, through influencing MD activity. A recent hypothesis proposes the basal ganglia implementing a fast reinforcement learning algorithm to discover MD activity patterns that improve current rewards [65]. In such a scheme, basal ganglia and possibly other cortical areas could exploit the ability to reconfigure the computations of another cortical area by simply modulating the associated MD, without requiring any long-term changes or plasticity. Future theoretical and computational work should be helpful in addressing these questions.

### Current and expected reward associations

Guided by fMRI findings, we examined how frontal cortical areas (dlPFC and OFC) might exploit the compressed representations in MD to coordinate their computations. The OFC is one of the main cortical targets of MD projections, and is engaged in goal-directed behavior [66]. Disconnecting OFC from thalamic afferent leads to impairments in assessing action-outcome associations [56]. The lateral parts of OFC, which we found in our fMRI analyses in the context of switching the strategy, represent predicted outcomes [67] and mediate the updated valuation of outcome desirability [68,69]. Another frontal region, the vmPFC encodes beliefs about outcome values [70] which can be an intermediate signal needed for the computations of belief over current context in OFC. One possibility is that vmPFC sends value information to MD where it can be read out by both dlPFC and OFC. However, since vmPFC was active in both arms of our fMRI contrast, our data did not make statements about where vmPFC might potentially send its value information, and we simply enter it as input to dlPFC. A cascade of frontal regions coordinating their computations seems plausible and will be the subject of future modeling work.

### Modular computational compartments

Our neural model used fixed thalamocortical connections and successfully captured behavior and flexible contextual switching, indicating a potential architectural design where the thalamocortical connections are either fixed or are updated at a much lower rate compared to corticothalamic connections. We initially reasoned that Hebbian learning at the thalamocortical synapses would induce formation of modules in the RNN, but in practice it created an echo chamber where the corticothalamic synapses learned an association and the thalamocortical synapses reinforced it, preventing the model from switching to other states, indicating that the brain might have special solutions for learning reciprocal projections between two areas.

Recent work by Tsuda and colleagues revealed interesting segregation of an RNN into distinct computational modules merely by diffusely scaling neuronal inputs multiplicatively, by neuromodulation effect [71]. In contrast, our neural model suggests contextual signals from MD that ultimately modulate activity of local cortical interneurons to disinhibit neuronal dendrites and scale their input gain [58]. Our setting allows for precise and more hierarchical reconfiguration of the RNN as each neuron gets a specific (yet randomly chosen) weight by which its inputs are scaled, creating multiple computational units or modules, defined by the domains of inhibitory interneurons. Physiological data report an MD multiplicative gating, and we here demonstrate its effect in partitioning the dlPFC activity into distinct

computational units, which is of great interest to understanding the neural basis of higher cognition on one end and developing machine learning counterparts on another. Segregating neural networks into modules might allow for flexible reconfiguration of frontal area circuitry and to solve tasks by composing modules in different combinations.

## Conclusions

Although primate MD is equipped with some unique features not identified in rodents, such as an intrinsic population of interneurons releasing GABA [72], our rodent MD-dlPFC neural model successfully captured contextual switching behavior in humans, suggesting a context-specific, evolutionary well-preserved interaction between both areas. Connectivity between MD and prefrontal cortex is however not restricted to single areas, but multifaceted so that individual neurons from different MD subdivisions exhibit reciprocal projections that diverge to simultaneously contact several different PFC subdivisions [30,73,74]. Distinct MD populations thus seem to facilitate corticocortical communication via trans-thalamic pathways [75] to support several different cognitive entities in rodents and non-human primates, such as working memory [76] and attentional control [20]. Whether the underlying MD-PFC interactions can also be translated to humans and segregated in a comparable way as shown by our MD-dlPFC neural modeling approach remains an open question for future studies.

Given that the MD appears to make cortical computations within and across areas more efficient, we imagine that perturbations of this process may decrease the computational efficiency of frontal network and result in possible behavioral abnormalities as a result (e.g. increased switching failures may manifest as perseverative behavior or obsessive thoughts in autism or obsessive compulsive disorder). More importantly, future work targeting the thalamus with non-invasive neuromodulation, guided by theory and models, could potentially provide a viable strategy for augmenting task-engagement, switching and general cognitive abilities in disorders of the frontal cortex.

## Methods

### Ethics statement

This study was approved by the ethics committee of the medical faculty at the Ruhr University Bochum, Germany (registration number 16–5786). All participants gave written informed consent prior to participation.

Code generated for this study has been deposited at <GitHub.com/hummosa/MD-reservoir>. Datasets generated during this study were previously published [32] and are available at < https://ruhr-uni-bochum.sciebo.de/s/xKBPyW7ZGLs2q2g>.

### Participant details

This study uses the previously collected dataset from twenty-eight healthy human participants (mean age ± SD: 25.3 ± 3.9 years) [32,36]. Only male participants were included to avoid influences of hormonal fluctuations over the menstrual cycle in females on learning and associated blood-oxygen-level-dependent (BOLD) signals [77–79]. All participants were right-handed as assessed by the Edinburgh Handedness Inventory [80] and had normal or corrected to normal vision.

### Probabilistic inference task design and structure

Participants were instructed to infer the next tactile stimulus from current cue by either choosing the same direction (match) or the opposite (non-match), and the predictability of the cue was manipulated by changing the strength of the cue-target contingency over time. The whole

experiment consisted of 10 blocks, 2 blocks for each of the five cue-target contingencies (i.e., strongly predictive: 90% or 10%; moderately predictive: 70% or 30%; and non-predictive trials: 50%). Each block consisted of an equal number of the two tactile patterns ('Up' or 'Down', presented for 500ms in random order as either cue or target stimulus. The sequence of blocks was pseudorandomized and fixed across participants to ensure inter-subject comparability [81,82]. Participants were informed that the cue-target contingency would change over time, but the exact probabilities were kept unknown. To avoid the prediction of a new block onset, the two blocks for each prediction strength were once presented with 30 trials and the other time with 40 trials. The experiment consisted of 350 trials in total, which we split into three runs, each lasting ~10 min. The details of the equipment used were described previously [32,36].

In the computational model, up and down cue inputs along with vmPFC input were fed to the dlPFC RNN using four input nodes. Trials were 200 time-steps long (to represent 200ms), and inputs were presented for the first 100ms. To directly compare human and model performance in Fig 1, the model required two pre-training blocks to match human performance at the beginning of the experiment, and also required 10 times as many trials in a block to reach the same performance at the end of the block. In all other analysis, model results are shown without any pre-training and the model was trained on blocks with 500 trials.

## FMRI data processing

The fMRI data acquired from 28 human participants in the previously published papers [32,36] were used in this study. The preprocessed data were re-analyzed using the general linear model (GLM) in SPM 12. Simply, images were applied to slice time correction, spatial realignment, and normalization to the MNI template using the unified segmentation approach [83]. Finally, normalized images were spatially smoothed using a Gaussian filter with a full-width half-maximum kernel of 6 mm. Data were high pass filtered at 1/128 Hz.

For each participant, we conducted a first level GLM. Events were time-locked to the onset of the presentation of the cue stimulus using stick functions and split into two regressors, one for *Staying* (no strategy switches) and the other one for *Switching* trials. Onsets were convolved with the canonical hemodynamic response function in an event-related fashion. Functional data from the three runs were concatenated using the spm_fmri_concatenate.m function in SPM12. Using this function, the high-pass filtering and temporal nonsphericity calculations were corrected to account for the original session length. Regressors of no interest included the presentation of the target stimuli (all trials collapsed to a single regressor), invalid trials (i.e., missing or late responses) and the six head-motion parameters as estimated during the realignment procedure.

Using the GLM, we investigated neural activity underlying switching the response strategy using the contrast "*Switching> Staying*". Based on the previous mice studies and to draw direct parallels to our neural model, we applied a smaller smoothing kernel (specifically, from using a 8mm smoothing kernel to a 6mm one), and we constrained the GLM analysis in the prefrontal areas and thalamus. Therefore, the resulting contrast images for all participants were applied to the group level one-sample t-test and thresholded at $p<0.05$, small-volume corrected for multiple comparisons at the voxel level (family-wise error rate, FWE) using anatomical regions of interest of the entire prefrontal cortex and thalamus (PFC was defined using the Automatic Anatomical Labeling atlas [84] and thalamus using Anatomy Toolbox [85]).

## Dynamic causal modeling (DCM) of fMRI data

To investigate effective connectivity and compare different network theories, we performed bilinear deterministic DCM [86] using SPM12. Following our GLM results and a-priori-

hypotheses, network nodes were represented by the two prefrontal regions, i.e., right dlPFC, OFC, and the thalamic MD in which neural responses related to *Switching>Staying* in the GLM. Subject-specific time series were extracted from the nearest local maximum within a sphere with a radius of 8 mm centered on each node's group maximum. The first Eigenvariate was extracted across all voxels surviving p = 0.05, uncorrected, within a 4 mm sphere centered on the individual peak voxel. The resulting BOLD time series were adjusted for effects of no interest (e.g., invalid trials, and movement parameters). Following these procedures, time series for all three areas could be extracted in 24 out of the 28 participants. In four participants we could not obtain right thalamic time series because activations did not meet the above criteria. These four participants were excluded from DCM analyses.

DCMs are specified in terms of fixed (endogenous) connections between brain areas and condition-specific changes in the strength of these connections (i.e., modulatory or bilinear effects). Given that every brain region is connected reciprocally [87], we assumed reciprocal endogenous connections between the three regions of interest. We specified models with different modulatory (bilinear) effects by the conditions (*Staying* and *Switching*). Specifically, we tested whether the feedforward, feedback or both connections between thalamus and dlPFC or OFC were modulated by *Switching>Staying*. To restrict model space, we fixed connections in both directions between dlPFC and MD, i.e., original mouse network switching motif [18]. This resulted in 6 competing models that were further evaluated (S5 Fig).

We used a two-step fixed-effects Bayesian model selection (BMS) to assess the most likely among a set of competing models about the mechanisms that generated observed data in dlPFC, OFC, and MD. The fixed-effect analysis was used as we assumed that all participants were best described by the same brain network, but with different connection strengths. In a first step, we used family-level inference to determine whether models with sensory input into dlPFC, OFC or MD best explained the observed data. For simplicity, we did not include lower-level sensory areas in our DCMs, focusing instead on the subgraph modeling of the brain regions engaged by switching strategy. Second, the models of the winning family were compared to identify the most plausible DCM of modulation effects by the strategy switching in the task.

BMS rests on the model evidence P(y|m), i.e. the probability of observing the data y given a particular model m, and uses the group log-evidence to quantify the relative goodness of models. The log-evidence of a model is calculated as the negative variational free energy under the Laplace approximation. It represents a generic tradeoff between the accuracy and complexity of a model that can be derived from first principles of probability theory. The sum of the log-evidences over participants (which is equivalent to the log group Bayes factor) can be used to compute how likely it is that a specific model generated the data at the group level (i.e., the expected model posterior probability).

Parameters of the winning model were then summarized by Bayesian parameter averaging (BPA), which computes a joint posterior density for the entire group by combining the individual posterior densities [88,89]. A posterior probability criterion of 95% was considered to reflect significant effective connectivity.

## Thalamocortical neural model

The model has four main structures: dlPFC as a recurrent neural network (RNN) with reservoir dynamics, MD as 2-neuron network with winner take all dynamics, vmPFC as an estimator of available action strategy values, and OFC as a Bayesian observer to estimate probability of an experimental block changepoint.

The dlPFC receives cue information input (up or down) along with estimated behavioral strategy rewards from the vmPFC (see 'Maximum Likelihood-Algorithmic model of vmPFC'

below). Network output was a readout projection from dlPFC to 2 neurons representing responses 'up' or 'down'. Reciprocal connections connect dlPFC and MD with weights to MD following Hebbian plasticity and weights from MD fixed with an additive and multiplicative effects on dlPFC. Two sets of weights were plastic: from dlPFC to output using node perturbation, and from dlPFC to MD using Hebbian learning. Connections from input to dlPFC, recurrent connections in dlPFC, and connections from MD to dlPFC are all fixed.

We describe each element below.

## Dorsolateral prefrontal cortex model architecture

We modeled the dorsolateral prefrontal cortex (dlPFC) as a reservoir RNN of 500 neurons with fixed recurrent connectivity weights $w_{ij}$. Input to each neuron $I_i$ evolved dynamically according to:

$$\tau \frac{dI_i(t)}{dt} = -I_i + \sum_k w_{ik}^{inp} Inp(t) + \sum_j w_{ij} r_j(t) + \sum_j \left( \sum_m w_{im}^{MD} r_m^{MD}(t) \right) w_{ij} r_j(t) + \sum_m w_{im}^{MD} r_m^{MD}(t) + \rho_i^{noise}$$

Where $w_{i,k}{}^{inp}$ were the weights from inputs unit to dlPFC neurons (described below). $w_{i,k}$ were the recurrent weights drawn from a gaussian (mean:0, var: 0.75/sqrt(2*Nsub), where Nsub = 200 is the no of cells receiving each input), with each row then zero-centered to maintain stability. $w_{i,k}{}^{MD}$ were the weights from MD. The time-constant t was set to 0.02 and the discretization time step dt was 0.001 ms. $\rho_i^{noise}$ is the noise added to each neuron generated as:

$$\rho_i^{noise} \sim \mathcal{N}(0, 1^{-3})(t)$$

Input was then passed through a tanh activation function and activations were clipped to positive values.

$$r_i(t) = [tanh(I_i(t))]^+$$

The multiplicative and the additive input from MD to dlPFC uses the same set of fixed weights drawn from a normal distribution of mean 0, and variance 0.1.

## Inputs to the dlPFC

Each cue input unit projected to a population of 200 dlPFC neurons, and each half of these 200 neurons received inputs from one of the value input units, i.e., strategy value (match or non-match) and cue (up or down) combinations are selectively projected to neurons in the dlPFC reservoir such that each strategy-cue combination projects to 100 dlPFC neurons. Weights from input units to their target dlPFC population were sampled uniformly with values between 0.2–0.4. Although inputs to the model were structured, the recurrent dynamics eventually dictated neuronal behavior (not shown), and rather we resorted to decoding neuronal responses using logistic regression as described below.

## Medio-dorsal thalamus model architecture

The medio-dorsal thalamus (MD) was modeled as two neurons with no recurrent connectivity but with winner-take-all (WTA) dynamics capturing inhibitory interactions with thalamic reticular nucleus. The two neurons received input from dlPFC neurons through Hebbian plastic weights $w_{ij}{}^{dlPFC}$. The two MD neurons had the same time-constant as dlPFC neurons and at each time step the neuron with the higher inputs had its activation set to 1 while the other had 0.

Learning at the dlPFC to MD weights followed a Hebbian rule, with the pre-synaptic eligibility trace evolving dynamically with a time constant $\tau_{pre}$ (2000ms), as follows:

$$\Delta\rho(t) = \frac{1}{\tau_{pre}}[r(t) - \rho(t-1)]$$

$$\Delta w^{dlPFC-MD} = \alpha\, \rho(t) r^{MD}(t)$$

Where $\Delta\rho(t)$ is the change in the Hebbian eligibility trace at time t, $r(t)$ and $r^{MD}(t)$ are the firing rates of dlPFC and MD respectively, and $\alpha$ is the learning rate (set to $5\times10^{-5}$). Both pre- and postsynaptic activities were centered around zero by subtracting their respective means. To prevent Hebbian learning instabilities, weights were clipped to [-0.1, 0.1] and rescaled to keep their L2 norm constant and the end of each trial. In experiments where MD was lesioned, we removed the multiplicative and the additive MD input to dlPFC neurons, but model performance was impaired due to a decrease in dlPFC activations, so we compensated by multiplying the recurrent connections in dlPFC with a compensation factor (1.3) to bring activation levels back up to values similar to the MD intact model.

## Output and readout weights learning

Learning at these synapses is implemented through node perturbation, a biologically-realistic approximation of backpropagation of error [40]. Node perturbation injects a small input into the output neurons and evaluates the changes in the network performance measure (e.g., reward or error). Each input synapse to that node is then changed by a product of the brief noise magnitude and activity of the pre-synaptic neuron, scaled by the improvement of network performance. Mathematically, input to the output neuron $i$ at time step $t$ evolves according to:

$$\tau\frac{dI_i(t)}{dt} = \sum_j w^{out}_{ij}\, r_j(t) + \zeta_i(t)$$

where $\zeta_i(t)$ is drawn from a uniform distribution between (-1, 1).

Weights $w^{out}_{ij}$ are then updated according to:

$$w^{out}_{ij} = w^{out}_{ij} + \mu\zeta_i(t)r_j(t)\delta e$$

where $\mu$ is the learning rate ($5\times10^{-5}$), $r_j$ is the activity of the pre-synaptic dlPFC neurons, and $\delta e$ is the change in reward received.

## Maximum likelihood-algorithmic model of vmPFC

We computed the likelihood of strategy values (or equivalently, association levels) over the range [0, 1] given strategies executed and rewards collected over a horizon of 10 recent trials. We calculate the expected reward probability from executing a match strategy (q) as:

$$P(q|d_{T-h:T}) = \prod_{t=T-h}^{T} P(d_t|q)P(d_t)$$

Where h is the number of trials in the horizon, $d_{t-h:t}$ are the rewards and actions in the horizon buffer, and T is the total number of trials. Strategy values sent to dlPFC are expected reward of match and non-match strategies outputted as {q, 1- q}.

We used the value with the maximum likelihood estimate as the current estimate of the q value and used it as output to dlPFC for the next trial. If the current estimate starts to diverge

from recent average of rewards by more than a threshold (0.15), the model considers a possible block change, and adds its current estimate to a stored bank of previously encountered values, and then considers if the recent average of rewards matches any of the stored previously encountered values, and if not, it concludes a newly encountered behavioral contexts and it creates a new estimate initialized at 0.5 and adapts it to match the recent average of rewards. As such, this model relies on slowly changing estimates of outcome values to disambiguate outcome values in different contexts. For a more rapid detection of context changes we also utilize a Bayesian model as described next.

## Bayesian observer representing OFC computation

The representation of latent variables and appropriately updating them at changepoints was modelled as a Bayesian estimate of the probability of a context switch, and two latent variables representing match or non-match context that get updated when the probability of a context switch reached a threshold. Specifically, the probability of a switch in context at some point in the most recent h trials as:

$$P(s|d_{T-h:T}) = \frac{1}{h} \sum_{t=T-h}^{T} \frac{P(d|s_t)P(s_t)}{P(d)}$$

Where $P(s|d_{T-h:T})$ is the probability of a context switch given the actions and outcomes over the last h (= 10) trials. The prior $P(s_t)$ is initialized at 1/T. $P(d|s_t)$ is the likelihood function of a switch at trial t considering the h most recent actions and outcomes ($d$), calculated as follows:

$$P(d|s_i) = \prod_{t=T-h}^{T-h+i} P(d_t|c) \prod_{t=T-h+i}^{T} P(d_t|\bar{c})$$

Where c is the current context belief nodes taking value [1,0] for match context and [0,1] for non-match context. They are modeled as a Markov chain simplified and made deterministic. Once the probability of a changepoint reaches threshold, the two context belief nodes are flipped and the buffer containing recent trials is cleared (horizon reset to current trial).

$$c = \begin{cases} c, P(s) < \theta_s \\ [1,1] - c, P(s) > \theta_s \end{cases}$$

The two nodes representing current inferred context $c$ are associated through Hebbian plasticity to active MD and dlPFC cells in that trial as follows:

$$\Delta w^{OFC-MD} = \alpha \, c \, r^{MD}$$

$$\Delta w^{OFC-dlPFC} = \alpha \, c \, r^{dlPFC}$$

Where $r^{MD}$ and $r^{dlPFC}$ were the trial-averaged-activity of MD and dlPFC neurons respectively, and $\alpha$ is the learning rate set at 0.001. Vectors c, $r^{MD}$, and $r^{dlPFC}$ were zero-centered. After each trial the weights from OFC to MD and to dlPFC were normalized to a length of 1 and 2, respectively, as OFC to dlPFC weights were shared amongst a larger number of neurons. The connections from OFC to the target areas were activated after the first two blocks of the experiment, allowing Hebbian learning to converge before producing behavior. They were also activated briefly in the beginning of each trial and we parametrized this duration to look for the shortest signal necessary to influence the dlPFC-MD circuit behavior (Fig 5D). In the experiment reported in Fig 5C, we varied the number of neurons receiving the signal we used

a duration of 10ms, conversely, in the experiment in e Fig 5D we varied the duration and used 500 as the number of neurons.

## Finding cue-responsive neurons with logistic regression

To find cue-responsive neurons, we applied logistic regression over the mean trial activity in the interval 50–150 time steps. For an individual neuron, we plotted the points (x, y) where x is the mean trial activity and y is 1 if the cue was up and 0 if the cue was down. Applying logistic regression gave us a pseudo r-squared value for each neuron; in the case that the data was perfectly linearly separable, we set r-squared to be 1.

Applying this technique to trials within the match and non-match contexts separately, allowed us to identify which neurons were cue responsive in one or both contexts, or not at all. These classes of interest correspond to the corners of the $R^2$ histograms in S3A Fig. Neurons that did not respond to cues were in the upper-left corner, and neurons that responded in both contexts were in the lower-right corner, whereas neurons that were cue responsive only in one of the contexts where in the upper-right and lower-left corners.

## Statistics

*Fig 1D*. To compare human and model performance across association levels, we compared the average performance per association level across all human participants (N = 28; mean age ± SD: 25.1 ± 3.8 years; male) and model simulations (N = 5). Bars represent mean +/- SEM; the mean represents average performance for high (90/10%, human mean 0.79 +/- 0.05, model mean 0.80 +/- 0.02), low (70/30%, human mean 0.58 +/- 0.05, model mean 0.59 +/- 0.02), and non-informative (50%, human mean 0.433 +/- 0.05, model mean 0.50 +/- 0.01) association levels, where an observation is the accuracy of a single trial (0 or 1 for correct or incorrect response. We performed a two-way ANOVA with multiple comparison testing (anovan. m, multicompare.m MATLAB) with high, low, and non-informative association levels as factors and human versus model interactions. Statistical significance was found across association level groups in both the human and model data (human high/low, $p < 0.0001$, high/non-informative, $p < 0.0001$, low/non-informative, $p < 0.0001$; model high/low, $p < 0.0001$, high/non-informative, $p < 0.0001$, low/non-informative, $p = 0.039$). Lack of statistical significance was found between human and model data for each association level grouping (90/10% and 70/30%, $p = 0.990$; 90/10% and 50%, $p = 0.990$; 70/30% and 50%, $p = 0.101$).

*Fig 1E*. In order to measure the correlation between human and model performance, we compared the average performance across all human participants (N = 28; mean age ± SD: 25.1 ± 3.8 years; male) to model performance (N = 5). We binned trial data across time with bin size equal to 15 and 150 for the human and model, respectively, for the same number of bins across human model data. We computed the mean of all bins, comparing the human and model mean per bin. The correlation coefficient r was calculated using corr2.m (MATLAB), and significance was determined by Pearson correlation coefficient using corrcoef.m (MATLAB). The correlation between human and model performance was Pearson correlation coefficient, r = 0.94, p = 1.35e-08.

*Fig 2A*. To investigate the neural activations underlying switching the response strategy, we compared the trials between switching and staying response strategy (*Switching* vs. *Staying*). Individual contrast images (Switching > Staying) were applied to the group level one-sample t-test in SPM. The group-level significant was thresholded at p<0.05, small-volume corrected for multiple comparisons at the voxel level (family-wise error rate, FWE) using anatomical regions of interest in the prefrontal cortex and thalamus.

*Fig 3E*. We used linear regression model to decode cue or context from either MD or dlPFC population trial-averaged activity and compared the accuracy of either model to chance

level prediction (50%). We tested 10 network instantiations and found that context was decodable above chance from both dlPFC or MD activity (p-values 3.7e-7, 1.5e-14) while cue was decodable from dlPFC activity (p-value 1.2e-8) but not MD activity (p-value 0.1778).

*Fig 3G and 3H*. To compare the mean accuracy of the MD lesioned and MD intact models we simulated 20 runs of the model with different initializations of the network (different random seeds). We averaged performances in each experimental block for each of the 20 runs. We used a t-test to compare the two groups of means from the MD lesioned and MD intact models in each block. P-values < 0.05 labeled with *, <0.01 **, < 0.001 with ***, >0.05 with ns.

*Fig 4A*. To compare the number of cue-responsive neurons in a model with and without a MD component, we identified and counted the neurons that were cue-response exclusively in match contexts, the neurons that were cue-response exclusively in non-match contexts, and the neurons that were cue-responsive regardless of the context across multiple simulations (N = 10; *see Finding cue-responsive neurons with logistic regression*). Bars represent mean +/- SEM; the mean represents average number of neurons for a given model and given cue-responsivity (match in MD model 37.5+/-2.57; match in MD lesioned model 52.3+/-3.58; non-match in MD intact model 39.3+/-6.10; non-match in MD lesioned model 55.2+/-2.63; context indifferent in MD intact model 71.4+/-6.55; context indifferent in MD lesioned model 138.9 +/-9.19). Pairwise comparison was computed within groups. Statistical significance was determined by a two-tailed Mann-Whitney test (match context p = 0.01; context-indifferent p = 0.9e-4). Lack of statistical significance in the non-match context was determined by a two-tailed Mann-Whitney test (p = 0.10).

*Fig 4B*. To examine which weights discriminated between the two models, we averaged weights from all cells identified as encoding cues only in a match context and grouped by whether they projected to the matching or the non-matching output neuron, and plotted their evolution throughout the experiment using alternative 90% match and 10% match blocks. We repeated the same plot for cue-responsive cells in both contexts. Shaded areas are +/- SEM.

*Fig 5E–5F*. We compared the effective connectivity within the winning model of DCM between *Switching* and *Staying*. Parameters of the winning model were summarized by Bayesian parameter averaging, which computes a joint posterior density for the entire group by combining the individual posterior densities. The parameter estimates for *Switching* and *Staying* trials were then contrasted against each other. A posterior probability criterion of 90% was considered to reflect significant difference. The parameters were significantly higher for *Switching* than for *Staying* trials for dlPFC->OFC, OFC->MD, MD->dlPFC and dlPFC->MD connections.

*S1F Fig*. We compared the prior belief–related activity in vmPFC between *Switching* and *Staying*. The prior belief-related activity (beta value) in vmPFC extracted from both *Staying* and *Switching* were applied to a two-tailed paired-sample t-test. The significant level defined as p < 0.05. Bars represent mean +/- SEM.

## Supporting information

**S1 Table. Brain regions related to the Strategy Switching (*Switching* > *Staying*) (p < 0.001, uncorrected).**
(DOCX)

**S2 Table. Brain regions correlating with the prior belief in *Switching* and *Staying*.**
(p < 0.001, uncorrected).
(DOCX)

**S1 Fig. Value inputs improve model behavioral flexibility and correlate with fMRI activity in human vmPFC.** We compared the model with and without value input (+vmPFC, -vmPFC

respectively), and also with and without output from MD (+MD, -MD, respectively). A. Performance of model with no value inputs shows little behavioral flexibility with significant dips in performance at block changepoints. B. Trial-averaged MD activity correlation with ground truth present context for each trial. Only model of both value inputs and intact MD output to dlPFC showed appropriate encoding of context in MD. C. The weights from dlPFC neurons to output showed significant learning and unlearning in model without value input. Adding value inputs leads to more coherent weight changes across blocks, and adding MD further reduce destructive learning and unlearning. D. We considered the distribution of output weights at the end of experiment and looked at its variance as a measure of dispersion. Weights that learned and then unlearned across blocks remained close to zero with low variance. E. The comparison of prior belief–related fMRI activity in vmPFC between *Switching* and *Staying* in human participants. Prior belief about the outcome value, derived from Hierarchical Gaussian Filter model (details in previously published papers [32,37]), correlated with the activity in vmPFC for both *Staying* and *Switching* strategy. The results projected on axial MRI brain slices. Brain activations displayed at p < 0.001 (uncorrected, red). For other regions see S2 Table. F. The prior belief-related activity (beta value) in vmPFC extracted from both *Staying* and *Switching* were applied to a paired sample t-test. The result showed no significant difference between *Staying* and *Switching* (p = 0.30). The error bar depicts the standard error.
(TIF)

**S2 Fig. Reponses of output neurons and MD neurons.** A. Responses of the 'Down' output neurons during a trial with input sensory cue as 'Up' or 'Down', first in a match context (left), then in a non-match context (middle), and trial-averaged activity of the same output neuron over the first three block of the experiment (right). Output neuron activity to the correct responses separates with readout weight learning over the first three blocks of the experiment and correctly reads out target output from dlPFC activity. B. Same as in A but for the 'Up' output neuron. C. Behavioral responses of the model across an experiment starting with 90% and 10% blocks, but then including a 70% and a 50% association level blocks. D. Responses of one of the MD neurons with some increased responses in the opposite context when the association level is less predictive. E. Weights from dlPFC neurons to one MD neuron, averaged over the 5 neurons with the highest eligibility trace in block 1 (blue), or the 5 lowest (orange) showing opposing weight dynamics across blocks.
(TIF)

**S3 Fig. Changes in representation and behavior in the MD lesioned model.** A. Histogram of correlation values (pseudo R-squared) between individual neuronal activity and input cue, with trials drawn from a match context on the x axis, and a non-match context on the y-axis (See Methods, 'Finding cue-responsive neurons with logistic regression'). The histogram for the MD intact model (left) and MD lesioned model (middle) were subtracted to highlight the differences (right), showing mainly fewer cells that correlated with input cue in both contexts.
(TIF)

**S4 Fig. Multiplicative thalamocortical projections separate context representations without obfuscating sensory cue representations.** Experiments with alternating blocks of 10% and 90% of match trials rewarded with no vmPFC inputs, but one MD neuron artificially activated for each type of block and Hebbian learning at the corticothalamic projections disabled. We tested the model with multiplicative or additive thalamocortical projections separated and parametrized the strength of either projections by multiplying their respective values by a factor from 1 to 40. A. Model performance for selected strengths of additive projections and B. multiplicative projections. C. Increasing the strength of additive projections initially improves

performance and behavioral flexibility at block changepoints but performance rapidly peaks and declines. D. Increasing the strength of multiplicative projections consistently increases performance until reaching 0.9 ratio correct steadily with minimal dips at block changepoints. E. Increasing additive projections strength decreased neural activity correlation in dlPFC for match vs non-match contexts, but also rapidly increased correlation between up and down trials until they become highly correlated, and presumably difficulty to decode. F. Increasing multiplicative project strengths decreases neural activity correlation in dlPFC between contexts with limited increase in correlation between cues.
(TIF)

**S5 Fig. Causal pattern space of DCM for fMRI data.** A. Illustration of the model space for Bayesian model selection. We specified six models to determine whether the feedforward, feedback or both connections between OFC were modulated by the strategy switching. We constrained the space to models with assumed dlPFC to MD reciprocal connections as the role of these connections have been demonstrated in animal studies [21]. Bayesian model selection revealed that among the models with the tactile input directed to dlPFC, model 3 was superior to the other 5 models. B. The log-evidence and posterior probability for each of the 6 models.
(TIF)

## Author Contributions

**Conceptualization:** Ali Hummos, Bin A. Wang, Sabrina Drammis, Michael M. Halassa, Burkhard Pleger.

**Data curation:** Bin A. Wang, Burkhard Pleger.

**Formal analysis:** Ali Hummos, Bin A. Wang, Sabrina Drammis.

**Investigation:** Bin A. Wang, Michael M. Halassa, Burkhard Pleger.

**Methodology:** Ali Hummos, Bin A. Wang, Sabrina Drammis, Michael M. Halassa.

**Project administration:** Michael M. Halassa.

**Resources:** Michael M. Halassa, Burkhard Pleger.

**Software:** Sabrina Drammis, Burkhard Pleger.

**Supervision:** Michael M. Halassa, Burkhard Pleger.

**Visualization:** Ali Hummos, Bin A. Wang, Sabrina Drammis.

**Writing – review & editing:** Ali Hummos, Bin A. Wang, Sabrina Drammis, Burkhard Pleger.

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
