## [Decision Letter · Decision Letter 0]

17 May 2022

Dear Dr Hummos,

Thank you very much for submitting your manuscript "Thalamic regulation of frontal interactions in human cognitive flexibility" for consideration at PLOS Computational Biology.

As with all papers reviewed by the journal, your manuscript was reviewed by members of the editorial board and by several independent reviewers. In light of the reviews (below this email), we would like to invite the resubmission of a significantly-revised version that takes into account the reviewers' comments.

While we ask you to thoroughly address all remarks (this is a major revision; the paper will be sent back to the original reviewers), we would like to attract your attention to the following points:

Several reviewers pointed to the fact that some features and characteristics of the model were rather obscure (both its structure and its function - including how it reacted to the lesion experiments). We believe that clarifying this part will crucially improve the readability (and possible impact) of your paper.

Concerning fMRI analysis, R1 points to a lack of clarity concerning the region of interest selection (this is a major issue in fMRI), as well as insufficient reporting: both these aspects should be taken extremely seriously during the revisions. Finally, R2 makes a very interesting and relevant remark, that is that the current implementation of the analyses does not take into account the model predictions (activity in the different nodes). I believe this approach could provide important insights.

We cannot make any decision about publication until we have seen the revised manuscript and your response to the reviewers' comments. Your revised manuscript is also likely to be sent to reviewers for further evaluation.

Sincerely,

Stefano Palminteri

Associate Editor

PLOS Computational Biology

Daniele Marinazzo

Deputy Editor

PLOS Computational Biology

While we ask you to thoroughly address all remarks (this is a major revision; the paper will be sent back to the original reviewers), we would like to attract your attention to the following points:

Several reviewers pointed to the fact that some features and characteristics of the model were rather obscure (both its structure and its function - including how it reacted to the lesion experiments). We believe that clarifying this part will crucially improve the readability (and possible impact) of your paper.

Concerning fMRI analysis, R1 points to a lack of clarity concerning the region of interest selection (this is a major issue in fMRI), as well as insufficient reporting: both these aspects should be taken extremely seriously during the revisions. Finally, R2 makes a very interesting and relevant remark, that is that the current implementation of the analyses does not take into account the model predictions (activity in the different nodes). I believe this approach could provide important insights.

Reviewer's Responses to Questions

**Comments to the Authors:**

Reviewer #1: Summary of the research and overall impression

This study aims to translate to humans the past research lines carried out on the rodent model: the computational modeling of the role of the mediodorsal thalamus (MD) on prefrontal cortex interactions during decision making. This is ambitious as the human prefrontal cortex is significantly more complex with debated roles of its sub-territories in decision making, and the MD holds unique primate features not identified in rodents. The authors propose a decision-making task switching across different contingencies inspired by past work on rodent models (changing visual vs. auditory cuing by up vs. down tactile cuing). It allowed them to use the computational model developed in rodents, and expend it in humans. Nicely, the authors report the fMRI activity during their decision-making task, allowing them to identify the orbitofrontal cortex (OFC) and dorsolateral prefrontal cortex (dlPFC) activities to correlate with switching behavior during their task, and demonstrate that MD is engaged in feedback with the dlPFC, and transmit OFC inputs during strategy switching. Overall, this paper is an ambitious and convincing attempt to model the role of MD in prefrontal cortices interactions during the strategy switch in line with contingencies changes during a simple decision-making task in humans. Regarding the thalamus, studies on the role of the MD in humans are mostly lesion studies, which are often non-specific, with a large spectrum of effects on amnesia, dementia, autonomic functions, mood, and sleep/waking cycle. Thalamic nuclei are small and deep structures that are difficult to record with classical non-invasive brain imaging studies, and the few intracranial recording studies focused on memory formation. In this context, proposing a computational model of the role of MD in humans is ambitious, but it brings valuable insights to this largely unexplored domain of research. In particular, this study gives a role to MD that is not being a relay of sensory inputs. To my knowledge, this is the first direct evidence of such a role of thalamus nuclei in humans. Moreover, such models could help to understand the deficits induced by lesions or dysfunction of MD in humans.

To me, the outcomes of this study fit the ambition and impact expectations of PLOS Computational Biology. However, if the computation part of the study appears robust and convincing, some clarifications on the selection of the prefrontal regions seem necessary (see major issues below). A more exhaustive report of fMRI results is expected, and the course of the presentation should be rebuilt to show more clearly the unbiased identification of the prefrontal regions of interest.

Major issues

1. The course of the presentation is misleading. While the challenge of the prefrontal complexity in humans is well exposed since the beginning, the literature on sub-territories functions in decision making/task switching is too succinctly introduced and discussed. The dlPFC is given in the introduction as the region of interest for the model with few justifications. Actually, the dlPFC was not reported in the Change>Keep contrast for the same fMRI task in Burkhard Pleger et al., Cerebral Cortex, 2020. Authors should consider using the fMRI data as an unbiased first step of their study to label the entities of their model, and then more extensively discuss this result in line with the existing human literature.

2. In line with the previous comment, it is unclear if all cortical regions were explored or if the analysis was a-priori constrained to the dlPFC, OFC, and ventromedial prefrontal cortex (vmPFC). For example, it is surprising that activations related to Prior belief appear only in the vmPFC. The table with statistical effect for each cortical ROI is a must-have supplementary element in this study (tables for both Switching>Staying and Staying>Switching contrasts, and Prior belief).

Minor issues

1. The thalamic fMRI data is constrained to the MD, justified by the fact that previous studies in rodents focused on MD, and by the connectivity pattern of the MD. However, the effect in the other limbic thalamic nuclei would have been nice to have, knowing that poor evidence exists yet on the functional differences between those nuclei in humans.

2. The statistical difference between correct predictions in blocks with low predictability compared in comparison to blocks with unpredictability is not reported. The results from the ONE-way ANOVA should be shown in figure 1D. Actually, outcomes from a two-way ANOVA with interaction would be preferable with the type (Human/model) and association level (90-10/70-30/50%) as factors.

3. The authors did a significant and appreciable writing effort. However, they should consider doing an additional rereading effort of their manuscript including spelling, spacing, figure labeling (e.g. figure 1D and 1E and inverted between figure and text), and figures (e.g. missing error bar in figure 1D).

Reviewer #2: In this paper, Hummos and colleagues refine a previously developed neural model of an executive frontal-thalamic (MD) circuit based on mice work, and train it on a human decision-making task based on either a matching or non-matching rule on tactile percepts associated with variable contingencies. Both the nature of the rule and the contingencies can be parametrically varied to stimulate flexible responding. As a major take-home, the data suggest that the MD could act as a transthalamic hub to support indirect cortico-cortical communication in humans which may be required when flexible responding is warranted, as also supported by neuroimaging data. Altogether the paper is well written and of high interest for the field and even the broader audience as it effectively stimulates thoughts and proposes new insights to reveal the computational principles at play within thalamocortical circuits, with a possible anatomical relevance. I believe there are a few points that could developed or clarified to increase the value of the paper further.

1/ The neural model

Some additional descriptions or statements could be provided to better explain some crucial aspects of the model, especially for non-specialists. When considering the main features of the model, it is crucial to distinguish between what is driven by well-established facts from what is actually a working hypothesis. For example, the rationale that the MD exhibits “winner take-it-all” computational properties should be better explained. Is this based on empirical facts or a computational hypothesis? The reference provided l.148 based on anatomical work from Pinault does not seem to support this statement. There are only 2 neurons considered for the MD in the model so is this a limit of the current model or one of its feature to implement the “winner take-it-all” rule? In a similar way can we really make meaningful functional comparison between the 500 neurons dlPFC reservoir and the 2 neurons MD? See line 353-354 for one instance of the latter.

I was also a bit confused on the statement regarding the RNN which “did not have a mechanism to encode recent rewards” (the paragraph l.161-165). It is then said that the model is provided with inputs representing the value of each rule “computed over recent trials”. But if it is computed over recent trials what is then the difference from having a memory of recent rewards? Please clarify the difference.

The data illustrated in figure 2C need to be more comprehensively described and analyzed. What do top and bottom panels represent exactly, what is the difference between the two and how does this ultimately support the conclusion that dlPFC, not OFC or MD acts as input region? There is only one short mention in the main text l.207-208 which is insufficient to understand this and it is crucial for the next parts of the paper.

Finally, I think it would have been great to discuss a little further the extent to which the proposed model also has anatomical relevance. I think it is relatively well described that the OFC is the recipient from multiple sensory streams (e.g. classic neuroanatomy work from Cavada et al., Cerebral Cortex 2000), consistent with a major integrative role for this region. So I expect that some neuroanatomists would have expected that inputs may be conveyed from the OFC to the dlPFC rather than the opposite. This at least should deserve a more thorough discussion. Perhaps views from Barbas & colleagues could also be considered here to strengthen this aspect (e.g. García-Cabezas & Barbas, 2017). In direct line to this, the final paragraph l.358-366 is very short and do not bring much data for the reader to help consider the relevance of the proposed model (inputs reach dlPFC first, not OFC or MD). This part would benefit from being a little more scholar with clearer explanations of the data considered and the types of analyses that have been done.

2/ Perturbation experiment

The inclusion of a perturbation approach is nice and could be very informative to test some of the predictions of the model. In its present form however, it is difficult to have a clear functional understanding of MD deletion in the RNN. The data presented on figures 3F-I need to more thoroughly explained and analyzed. At present the main text and the caption do convey ambiguous statements. The main text says that the initial effect of MD lesion on flexible switching became less prominent after further training (l.233-234) but the caption of figure 3 (l.253-254) state that initial blocks showed no lesion effect unlike subsequent blocks which suggests the opposite. So I don’t really understand the nature of the deficit here. A transient deficit or acquisition impairment is a common feature in the rodent literature documenting the effect of MD lesion and has been relatively widely been discussed (there are multiple reviews out that document this specific issue) and it would be nice to compare that with the present dataset. I think the effect of MD lesions really need a much clearer description on what exactly is being impaired, what is not, and whether any of the effects are permanent or temporary. Given the scope of the paper, it would be expected to compare that with actual lesion data, there is a myriad of those available both in rodent and NHPs. This may help to strengthen the relevance of the model further and/or to identify some limits.

3/ Context vs rule

I was confused by some of the terminology used in the paper and in particular the use of the term ‘context’ which I did not expect to be necessary here. Both ‘temporal context’ and ‘behavioral context’ are used in the ms to describe the fact that the rule can change (matching vs non-matching). The first occurrence of this is line 218 and I really don’t understand the rationale for labelling this as ‘temporal context’ when what is really meant here is that MD neurons may encode the current rule within a block. Introducing ‘context’ here seems unnecessary and actually incorrect since there is no element to signal a context. Adding ‘temporal’ (and then ‘behavioral’ and multiple further instance) only makes the issue even worse and obscure much of the interpretation. I would really suggest to remove these terms throughout the whole ms or to better explain the rationale for using them (and especially what it brings to use those). To identify the current rule, participant must track any difference between expected and actual outcome but I really don’t see how the context could play a role in this.

I have a few more minor comments.

1. I think that the data considered in the introduction and the whole rationale to present the focus on the MD is a bit thin. There is a lot of literature out there and only 2 MD papers are considered. It should be easy to build a stronger case considering the vast literature available. Similarly, I found the initial question l.71 to introduce the present work a bit narrow as it only asks whether the MD could mimic Pulvinar functions on frontal systems. The Pulvinar surely is a high-order thalamic nucleus like the MD but unlike the MD it is modality specific so I think the MD is a candidate to have actually possibly broader (even more integrative) functions, at least across modalities. I think it would impact more significantly on the general audience to better introduce the present work and how it is positioned to influence what is now a quite dynamic field, with important functional questions.

2. The mentions on the non-relay functions of the MD may appear a bit trivial in 2022 (e.g. l.78, l.202, l.205, and especially l.209). Surely we knew it before the present work was conducted that the MD is not relaying tactile sensory information to the prefrontal cortex. Consider the sentence l. 208-209; alternatively, what would be the evidence demonstrating instead that a thalamic area actually engages in a task in humans to relay sensory information? I don’t think this bring much to main topic of the work and it is possibly distracting.

To get back to the previous point, I even wondered upon reading to this why the authors did not look at the Pom instead (the high-order thalamic nucleus for somatosensory information) as an actual parallel to the Pulvinar (especially l.205)?

3. Considering the model was initially developed to account for the performance of mice in a cross-modal attention task, I think it would be great to compare more clearly the mouse task with the human one as they have a number of clear differences which may impact at the functional level (e.g. cross-modal versus purely tactile task, cue signaling the current rule versus need to monitor the difference between expected and actual outcomes to identify the current rule, mice learn everything by trials-and-errors while participants are warned they will be rule changes – even though the changes are not directly signaled etc…).

4. In the discussion, the rationale for the statement l.437 that “thalamocortical connections are either fixed or updated at a much lower rate compared to corticothalamic projections” is not provided or explained. Please expand here. Is the fact that CT projections typically outnumber TC projections relevant here?

5. Some typos: l.139 perhaps a ‘to’ is missing, l.157 ‘disinhibitory’

Reviewer #3: The manuscript by Hummos and colleagues provides a modelling study investigating thalamocortical circuit mechanisms of context-dependent learning. This is combined with both behavioural and fMRI data from a human learning experiment, to which the model was applied. DCM on the fMRI data is used in attempt to arbitrate between different model architectures. In the task, participants are presented with a tactile stimulus (up or down) to which they have to respond with a match- or non-match response. The cue-response mappings (match- vs non-match) are probabilistic and change between blocks of trials (0.9, 0.7, 0.5 probability of the match- vs non-match response being correct).

The neural network model had four components: dlPFC (modelled as RNN with reservoir dynamics), mediodorsal thalamus (MD), vmPFC (providing value-related input), and OFC (providing estimates of the probability that the underlying latent state has changed). Importantly, thalamocortical connections (MD dlPFC) were fixed, while corticothalamic (dlPFCMD) weights undergoing Hebbian learning. The model behaviour closely tracks human choice data and model vmPFC output closely follows the underlying action probabilities. fMRI results show increased activity on strategy swith vs stay trials in dlPFC, MD, and OFC. DCM applied to these fMRI data supports a model in which sensory inputs first enter dlPFC, rather than being relayed via MD.

There are several results in this paper that I found very interesting. The authors investigated the coding properties of model dlPFC and MD neurons. In line with previous experimental findings, the architecture gave rise to mixed selectivity in dlPFC neurons. In contrast, MD neural activity was tightly related to the current context, or block (match vs non-match). Notably, the latter property of MD neurons relied on the long time scale of the Hebbian eligibility trace. When this time constant was reduced, MD neurons were biased to tracking the sensory signal (up vs down cue) that varies from trial to trial. Learning data of a model with MD lesions (output of MD to dlPFC removed) showed a "reversal learning deficit" - the lesioned model took longer to learn the new contingencies following a covert block switch, but it eventually reached control levels within a few hundred trials. The MD lesion also reduced cue selectivity in dlPFC neurons. Furthermore, in the intact model, connections between dlPFC reservoir to output neurons for the appropriate context increased for the correct context, but remained silent out of context. Finally, DCM results provided evidene in favour of transthalamic routing of OFC signals (OFC signalling to dlPFC via MD, rather than direct cortio-cortical signalling).

As stated above, I think this manuscript provides important new insights regarding the thalamocortical circuit mechanisms underlying context-dependent learning/probabilistic inference. I should point out that I am not an expert in neural network modelling (far from it), so I cannot provide a well-founded judgement on these methods. My comments below also mainly pertain to clarifying some questions that came up while reading the manuscript.

1) I think the cartoon of the model in figure 1B should be elaborated. The neural network model takes centre stage, so this needs to be clear. Before I arrived at the methods, it had not been really clear to me which components the model included - and which brain areas they were thought to represent. For example, I initially thought that the authors used vmFPC and OFC interchangeably, until I realized that these were separate components. Perhaps it would even be an option to have a separate figure solely dedicated to the model (and the variations of it that were used) to make the approach more straightforward to follow. Questions on the architecture that still remain open to me:

- How exactly does the MD component of the model look? I think that it contains two excitatory neurons that project to a population of inhibitory cells (hypothesized in the reticular nucleus) that provide feedback inhibition - is this correct?

- The OFC component is (at least what I understand) not at all present in the schematic (1B) - was it added only at a later stage?

- Is there anything about the RNN architecture that would make it particular to the dlPFC? In other words, could this component map to any cortical area (with strong recurrent connectivity)? Did the mapping of the component to dlPFC result from the fMRI findings?

2) I was intrigued when looking at the model MD neuron firing patterns (Figure 3D) - this looked to me like a code that very sharply represents the latent task state - thus holding a representation very similar to those found in OFC (Wilson et al., Schuck et al.). I have difficulty understanding how the representation found here would differ from an OFC representation of latent states. The MD representation seems to be very sharp (apparently emerging immediately after a block switch) and binary (where I would assume the latter part results from the winner-take-all dynamics in MD?). Could you please elaborate on how this emerges (in particular so abruptly) - is it a representation that is inherited from the OFC component, which I assume will be similarly sharp (threshold crossing of the switch probability)?

3) The fMRI results (with the exception of the DCM) are somewhat touched only quite briefly. In particular, I had been wondering what the rationale was for using a simple switch>stay contrast as the main fMRI contrast (which yielded dlPFC, OFC and MD; figure 2). I think this could be elaborated a bit in order to relate the regions more to their underlying functions as specified in the model. Wouldn't it be interesting to use e.g. the output of the OFC component as regressor in the GLM to test whether this region truly tracks the underlying latent state? Similarly with the model MD output (possibly also dlPFC). I appreciate that the dominant focus of this manuscript leans towards the modelling, but I would hope that a more thorough analyses of the fMRI data could further strengthen the authors' case.

Minor:

1) In the introduction, the authors sometimes switch between pulvinar and MD thalamus (e.g. line 71, 72) - would the authors see them as equivalent?

2) The task task figure (1A) does not seem to show the non-predictive (p = 0.5) context? Further to this, what do the eight circles represent? In figure 1C, it is not quite clear to me what is shown with the dots. I think more description in the caption could help. Overall, while I found all other figures rather straightforward to read and clear, I was quite much struggling with this one (see also my questions about model architecture above).

3) It seems that some literature is not cited correctly. In the introduction (line 56/57), two studies (Takahashi et al., Schultz et al.) are cited in support of OFC-dlPFC interactions, but as far as I remember neither of these investigated this topic.

4) I was wondering what specific purpose the HGF was used for? It only briefly comes up, almost in passing on page 5. It is mentioned that prior beliefs about outcome probability were weaker on switch compared to stay trials - and then the authors move on. This seems unnecessary to me, but maybe I'm missing something, in which case I would ask the authors to elaborate.

5) For the model MD lesion studies, blocks with 0.9 and 0.1 match probability were used - unlike main experiment with 0.9, 0.7, 0.5. 0.3, 0.1 - why?

**Have the authors made all data and (if applicable) computational code underlying the findings in their manuscript fully available?**

Reviewer #1: Yes

Reviewer #2: Yes

Reviewer #3: None

PLOS authors have the option to publish the peer review history of their article (what does this mean?). If published, this will include your full peer review and any attached files.

Reviewer #1: **Yes: **Antoine Collomb-Clerc

Reviewer #2: **Yes: **Mathieu Wolff

Reviewer #3: No
---

## [Decision Letter · Decision Letter 1]

19 Aug 2022

Dear Dr. Hummos,

We are pleased to inform you that your manuscript 'Thalamic regulation of frontal interactions in human cognitive flexibility' has been provisionally accepted for publication in PLOS Computational Biology.

Best regards,

Stefano Palminteri

Academic Editor

PLOS Computational Biology

Daniele Marinazzo

Section Editor

PLOS Computational Biology

Reviewer's Responses to Questions

**Comments to the Authors:**

Reviewer #1: The authors exhaustively answered the comments of the first review. The manuscript appears strengthened, with an increased impact. Once again, this study is of high interest to the field as it highlights a new role of the human thalamus, and may help the understanding of symptoms following thalamic lesions.

Importantly, major issues concerning ROIs have been well clarified in both the results and methods sections. The methodological approach to fMRI data is clearer in the text. It is now straightforward why results may differ from previously published work. Moreover, the effort of the authors to bring a more detailed introduction of human PFC subregions is appreciable for the conceptualization of the explored ROIs. Finally, constraining the analysis to specific ROIs based on a strong hypothesis related to previous studies seems fully acceptable, as soon as it is well specified, which is now the case.

Regarding minor issues, thank you to the authors for replying and modifying exhaustively in line with the comments. The first comment arose more from scientific curiosity, and the author’s answer suggests that switching/flexibility appears specific to MD. Other limbic nuclei are likely involved in this task, but more in memory/reinforcement learning processes.

Reviewer #2: The authors have addressed all my comments. I really appreciate the efforts and as a whole, the paper is clearly even more interesting to me. On the definition side of things, I am still wondering if a term like "state" (state representation) could be maybe better suited than "context" but this comment is only the expression of my scientific interest, I am not asking to discuss the issue further especially after the definition that the authors provided. I have no further comment.

Reviewer #3: The authors have clarified all my questions, performed a thorough revision and I think the manuscript is now even better and clearer to read. There is only one remaining point from my side. Previously, I had suggested to regress the model outputs against the fMRI signal. The authors have argued that this likely can't be done in a meaningful way, as the model outputs a 'generic average behaviour' - which does not account for the individual subjects' variation in when they switch their decision strategy. I would fully agree with this. But then I read in response to a comment by reviewer #2 that the RNN was provided with inputs representing recent rewards, and that these were experimenter-computed. This made me wonder whether it would not be possible to provide subject-specific value estimates to the RNN? Could one not fit a simple RL model to each subject's behaviour - this way the inputs to the RNN would take into consideration the individual sequence of choices made, and outcomes experienced by the participants. Please consider this merely a suggestion. If this could work the way I think, then maybe it might be worthwhile trying it. But to be clear: I think the paper is strong and comprehensive the way it is now - it does not require this extra analysis. Congrats on this exciting work.

**Have the authors made all data and (if applicable) computational code underlying the findings in their manuscript fully available?**

Reviewer #1: Yes

Reviewer #2: Yes

Reviewer #3: Yes

PLOS authors have the option to publish the peer review history of their article (what does this mean?). If published, this will include your full peer review and any attached files.

Reviewer #1: **Yes: **Antoine Collomb-Clerc

Reviewer #2: **Yes: **Mathieu Wolff

Reviewer #3: No

---

## [Editor Report · Acceptance letter]

6 Sep 2022

PCOMPBIOL-D-22-00338R1 

Thalamic regulation of frontal interactions in human cognitive flexibility

Dear Dr Hummos,

I am pleased to inform you that your manuscript has been formally accepted for publication in PLOS Computational Biology. Your manuscript is now with our production department and you will be notified of the publication date in due course.

With kind regards,

Zsofia Freund
